# In vivo pressure gradient heterogeneity increases flow contribution of small diameter vessels in grapevine

Martin Bouda [1,2]*, Carel W. Windt [3], Andrew J. McElrone [4,5] & Craig R. Brodersen [1]

Leaves lose approximately 400 $H_2O$ molecules for every 1 $CO_2$ gained during photosynthesis. Most long-distance water transport in plants, or xylem sap flow, serves to replace this water to prevent desiccation. Theory predicts that the largest vessels contribute disproportionately to overall sap flow because flow in pipe-like systems scales with the fourth power of radius. Here, we confront these theoretical flow predictions for a vessel network reconstructed from X-ray µCT imagery with in vivo flow MRI observations from the same sample of a first-year grapevine stem. Theoretical flow rate predictions based on vessel diameters are not supported. The heterogeneity of the vessel network gives rise to transverse pressure gradients that redirect flow from wide to narrow vessels, reducing the contribution of wide vessels to sap flow by 15% of the total. Our results call for an update of the current working model of the xylem to account for its heterogeneity.

[1] School of Forestry & Environmental Studies, Yale University, 195 Prospect St., New Haven, CT 06511, USA. [2] Institute of Botany of the Czech Academy of Sciences, Zámek 1, 25243 Průhonice, Czech Republic. [3] IBG-2: Plant Sciences, Forschungszentrum Jülich, Leo Brandt Straße 1, 52428 Jülich, Germany. [4] Department of Viticulture & Enology, University of California, 595 Hilgard Ln, Davis, CA 95616, USA. [5] USDA-ARS, Crops Pathology and Genetics Research Unit, Davis, CA, USA. *email: martin.bouda@ibot.cas.cz

Plants lose over 400 $H_2O$ molecules in exchange for a single molecule of $CO_2$ to fuel photosynthesis[1]. Large quantities of water must therefore be drawn from the soil and transported to the leaves to adequately support this exchange of atmospheric gases, and to prevent leaf desiccation. Xylem has evolved to accomplish long-distance water transport in plants with very little energetic cost once the initial investment in the xylem conduits is made, as it relies on the evaporation of $H_2O$ from the leaves to drive sap flow in response to the pressure gradient established by transpiration itself. Xylem acts as a porous medium, whose organization of void spaces (i.e. the volume within each xylem conduit) and resistances (e.g. the frictional resistance of the conduit walls, and the pit membranes that exist between adjacent conduits) determines how the pressure gradient propagates, and thus the rate and characteristics of sap flow to the leaves. While axially oriented and pipe-like in general, xylem conduits form a complex three-dimensional network embedded in an opaque matrix of both living and dead cell types, and are subject to strong negative liquid pressures, making direct observations of network structure and sap flow difficult. Despite the broadly accepted fundamentals of the Cohesion-Tension mechanism used to explain vascular plant water transport[2–5], the spatial distribution of pressure gradients and the resulting patterns of sap flow within the xylem network remain poorly understood.

Angiosperms transport most water through long, narrow conduits called vessels, each comprised of the continuous lumina (void spaces) of an axially oriented series of hollow cells that are dead at maturity. Flow between vessels is facilitated by perforated walls where vessels are adjacent, including at overlapping tapered vessel endings. Because the vessels are nearly circular in cross-section, water transport is often approximated with the Hagen–Poiseuille (HP) equation, which describes liquid flow rates in cylindrical pipes where volume flow rate is proportional to the radius raised to the 4th power, such that a small increase in radius leads to a large increase in flow. This framework shows good agreement with experimental data on the hydraulic conductivity of excised stem segments if vessels are cut open at both ends and a uniform pressure gradient is applied to the network of vessels[6]. For flow in actual xylem connected to both roots and leaves, maximum theoretical values based on HP predictions for an observed vessel diameter distribution must be reduced by empirical factors accounting for the resistance of the inter-conduit pit membranes embedded in the perforated end and side walls between adjacent vessels[2]. An open question of fundamental importance to our understanding of the biology of plants still remains unanswered: do these reductions apply uniformly, such that vessel networks still follow a bulk form of the HP relation between radius and flow rate in planta?

Given the difficulty in resolving the three-dimensional complexity of the vessel network and their connections, analyses of xylem flow commonly assume the pressure gradient within the network to be homogeneous[7–10]. The current working model of xylem (Fig. 1a) simplifies its structure to parallel files of cylindrical conduits, whose lengths are uniform or independently drawn from an identical distribution, and whose tapered end walls overlap serially[11]. Each idealized conduit consists of one lumen and one permeable end-wall placed in series, so that the total conduit resistance is simply the sum of the two resistance components, similar to Ohm's Law. Water is carried along each file of conduits by an axial pressure gradient, crossing successive pairs of lumen and wall. Transverse heterogeneity in structure or pressure gradient is not considered. The simplicity of this conceptualization has made it a powerful working model, which underpins our understanding of xylem hydraulic properties[11] as well as the methods we use to measure them[12,13].

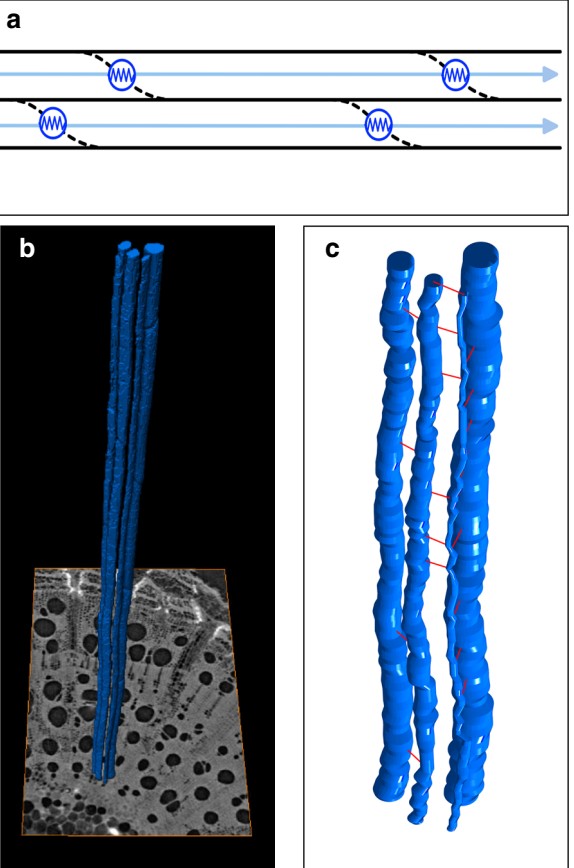

**Fig. 1 Idealizations of xylem anatomy vs. networks derived from X-ray micro-computed tomography (µCT). a** Current working model of xylem simplifies the xylem to parallel files of conduits of equal length, allowing it to be subdivided into units of one lumen and one end-wall resistance. Successive lumina are connected serially, with no connections across vessel files. **b** Four vessels (blue) segmented from a 3D stack of µCT images (at base). **c** Idealization of the same four vessels in the present model: each lumen is represented as circular in cross-section, with radius deduced from the data; connections (red) are placed at the centre of intervals where images show vessels to be connected.

While the homogeneous idealization allows us to approximate bulk xylem hydraulic properties[7,8,14], it cannot be used to determine properties of individual conduits, their mutual connections, and other details of the xylem network with important implications for xylem functioning. Such characteristics can strongly affect xylem hydraulic performance, including under stressful conditions that limit or impair flow. They have implications for the hypothesized trade-off between xylem hydraulic safety and efficiency, our understanding of xylem construction strategies[14], and optimal vessel diameter–length relationships for minimizing total xylem hydraulic resistivity[8]. Similarly, the model cannot account for effects arising from patterns of vessel network arrangement and connectivity, although these both affect the functioning of individual conduits and how vessel properties scale to bulk xylem hydraulics[15–17].

Real xylem is known to deviate from the homogeneous medium model in important ways at multiple scales. Lumen conductance is reduced from HP predictions due to variable conduit profile and radius[9] and the anatomy of connections between the cells that form a vessel[18]. Inter-vessel connections are not placed exclusively at vessel ends, so that water will traverse more end walls than would be predicted from vessel length alone. Vessel endings can be irregularly spaced[19] and concentrated in stem

nodes, especially in young stems[20]. The pattern of vessel connections commonly forms discrete clusters of conduits, aggregated in a non-random way[21], setting up parallel paths of diverse resistances. Vessel diameter, length, and connectivity can differ between xylem domains within a single cross-section[22], and the degree of integration or sectoriality across xylem domains is also variable, leading to varying levels of hydraulic isolation of plant parts[23–25]. Each such irregularity has the potential to significantly affect the overall xylem hydraulic function.

Although such component irregularities of xylem structure are observable, moving past the assumption of xylem homogeneity remains difficult because we lack observations that would enable us to resolve overall hydraulic function to their individual contributions. At the same time, simulations of flow through synthetic xylem networks have shown that non-random patterns of packing conduit clusters change overall tissue conductivity and vulnerability, emphasizing the value of basing such simulations on a full description of observed xylem networks[15,16,26]. Here, we combine two methods to address this longstanding knowledge gap. First, we measure in vivo flow in a transpiring grapevine (Vitis vinifera L.) stem with magnetic resonance velocity flow imaging (MRI). We then remove the segment of the stem visualized with MRI, and digitally extract the 3D network by scanning the segment with high-resolution X-ray microcomputed tomography (μCT, Fig. 1b, c). Flow through the xylem network is then modelled to determine what pressure gradients would need to exist for best agreement with the in vivo MRI data. Using these complementary imaging tools and model-data integration, we find high spatial heterogeneity of the pressure gradients that affect the flow of water through individual vessels or vessel groups. The observations imply reduced flow rates in the widest vessels, by 23% compared to theoretical predictions. As this represents a redirection of 15% of total flow towards narrow vessels, our results reveal xylem heterogeneity to be a key element in its construction and functioning.

## Results

**Imaging**. The clear positions of vessels and dilated phloem rays in both MRI maps of stem water content (Fig. 2a) and μCT scans (Fig. 2c) allowed us to coordinate the two spatially (Fig. 2e). In vivo flow rates in xylem conduits observed with MRI had values (mean ± standard deviation used throughout, unless otherwise noted) of $0.75 \pm 0.86\ \mu g\,s^{-1}$ ($n = 553$ pixels), with a median of only $0.44\ \mu g\,s^{-1}$ and a tail with 4.2% of flows above $3\ \mu g\,s^{-1}$, reaching as high as $5.3\ \mu g\,s^{-1}$ (Fig. 2b, Supplementary Fig. 1). Despite minor differences between the two scans (see Methods for details), the resulting image overlays of flow rates and vessel position (Fig. 2d, f) enabled us to approximate observed flow rates with modelled flow through the μCT-derived vessel network, at the scale of MRI pixels.

Analysis of μCT imagery of a 3.3 mm long segment from within the scanned region of the intact plant imaged with MRI yielded a total xylem cross-sectional area of $10.4\ mm^2$ in the stem with 491 separate vessels with diameter $37.5 \pm 29.3\ \mu m$. We were able to reconstruct the full vessel network within this 3.3 mm long stem segment, which consists of edges classified either as vessel lumina or inter-vessel pit membrane connections and the nodes where they connect. All vessels were open both at the upper and lower boundaries of the scanned volume, with no endings in between. They were connected by 485 separate connections, including 402 inter-vessel pit fields, with an average pit membrane area of $0.015 \pm 0.012\ mm^2$ each and 83 vessel relays[22] with an average diameter of $10 \pm 11\ \mu m$. The relays themselves included 204 relaying pit fields with an average connection area of $0.004 \pm 0.012\ mm^2$. The resulting network consisted of

317 separate vessel groups, made up of 199 solitary vessels, 189 vessels with a single connection (both pit field and vessel relay), and 103 with more than one.

**Transverse pressure gradients**. We modelled flows on the reconstructed μCT-derived network of resistances of vessel lumina and inter-vessel connections, minimizing the difference between modelled and observed MRI flows by adjusting pressures at the top of the sample. In a baseline scenario for comparative purposes, we imposed a uniform pressure at the top of the sample. The model was not able to fit observations with this boundary condition (Fig. 3). The best-fit uniform pressure drop across the stem, 0.055 kPa (or a gradient of $0.017\ MPa\,m^{-1}$), led the model to underpredict total flow by 25%. Flow was most strongly underpredicted in the inner xylem domain surrounding the pith, but simultaneously overpredicted in the outer domain (see Fig. 2c for domain delineation), so that the Nash–Sutcliffe Efficiency[27], or NSE (see Methods), of the model was only 0.31.

To eliminate the influence of the outer domain on results in the inner domain, we performed the baseline simulation again, with MRI pixels in the outer domain masked out. Fit inside the inner domain improved significantly (NSE = 0.79), but there was still a model bias, with 12% of the flow observed in the inner domain missing. Alternatively, if the total flow was fit instead of MRI pixel residuals, the bias was eliminated but NSE dropped to 0.76, with a best-fit pressure drop of 0.135 kPa (gradient of $0.041\ MPa\,m^{-1}$). In both cases, model overprediction clustered around wide vessels and underprediction clustered around narrow ones (Fig. 4), indicating that, in vivo, xylem characteristics redirect flows away from the widest vessels, as compared to HP predictions without transverse pressure gradients.

In the second scenario, pressures at the top of the sample were set independently in each of the 317 vessel groups. This allowed the predicted flow to match the observations more closely (Fig. 5). Total predicted flow matched the observation to within 0.78% and the model had a near-perfect NSE of 0.98, indicating that its spatial distribution was also very close to the observations. The pressure differences imposed across the sample in this second scenario ranged from 0 to −0.78 kPa (Fig. 6a, pressures weighted by flow and averaged for MRI pixels, see Methods). These model results show that given the reconstructed vessel network, the flow pattern observed in vivo implies strong transverse heterogeneity of pressure gradients.

The flow-weighted mean axial pressure gradient in the second scenario was $0.069 \pm 0.069\ MPa\,m^{-1}$, (Fig. 6b), which suggested a pressure drop of around 200 kPa across the entire 2.5 m long stem. This is broadly consistent with the plant's well-watered and illuminated status, especially considering that most hydraulic resistance resides in the roots and leaves of a plant. Transverse gradients between adjacent non-zero pixels ($n = 1273$) ranged from 0 to $9.1\ MPa\,m^{-1}$ and had a mean of $0.93\ MPa\,m^{-1}$ and a median value of $0.43\ MPa\,m^{-1}$ (Fig. 6c), making them about an order of magnitude stronger than the mean axial gradient. This result agrees with the hydrostatic principle that for steady flow in porous media, the strongest gradients will develop across layers with lowest conductivities[28], as well as with previous MRI flow observations that imply pressure discrepancies between vessels in a transverse plane[29]. As the conductivity of the xylem network is highest axially, gradients in this direction will locally be weakest, even though it is the dominant direction of overall flow.

**Spatial arrangement of xylem domains**. Flow was concentrated in the dorsal and ventral domains of the inner xylem domain (see Fig. 2c for domain delineation). This region contained only 35% of the vessels but accounted for 70% of the total flow. Vessels in

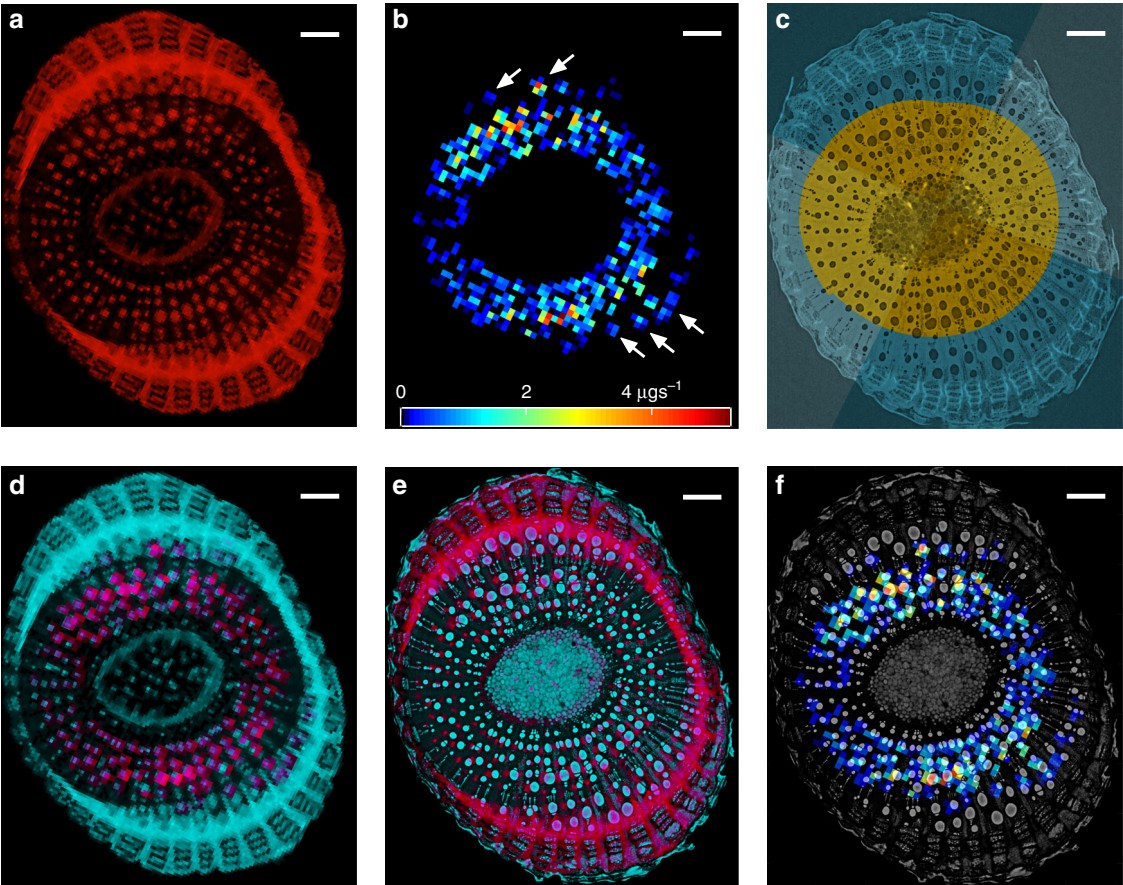

**Fig. 2 Stem scans. a** in vivo Magnetic Resonance Image (MRI) of a *Vitis vinifera* (L.) stem at 39 μm resolution, image intensity follows water content. **b** MRI of in vivo water flow rates in the same segment at 78 μm resolution; white arrows indicate significant flows in vessels of the outer domain. **c** X-ray micro-computed tomography (μCT) scan at 3.2 μm resolution of the excised internode, where xylem is divided into inner (yellow) and outer (blue) domains as well as Dorsal-Ventral (darker wedges) and Lateral (lighter wedges) domains. **d** Overlap of (**a**) (red) and (**b**) (cyan). **e** Alignment of a with μCT images: inverse μCT intensity in cyan, MRI intensity in red, overlap in purple. **f** projection of volume flow map (**b**) onto μCT image (greyscale). All scale bars are 0.5 mm long.

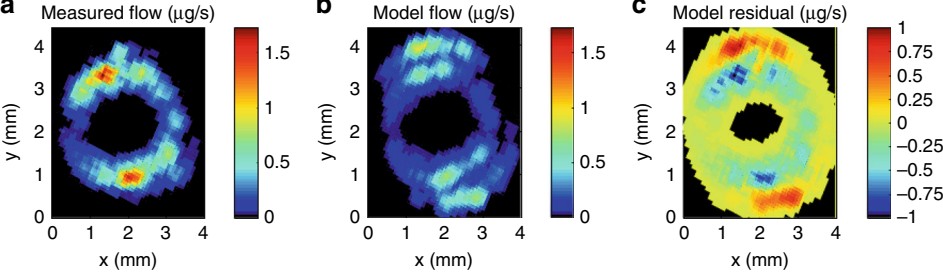

**Fig. 3 Baseline scenario. a** Observed flow rates (μg s$^{-1}$, colour scale) averaged in 5-by-5 pixel window. **b** Model predicted flow rates (μg s$^{-1}$, colour scale) with single pressure value set at top of domain, downsampled to 78 μm resolution and averaged over 5-by-5 pixel window. **c** Residuals between model and observed (μg s$^{-1}$, colour scale).

the inner domain ($n = 343$) had a mean axial pressure gradient of $0.06 \pm 0.11$ MPa m$^{-1}$, significantly higher ($P < 0.01$, $F = 23.45$, 2-way analysis of variance or ANOVA) than the outer domain ($0.02$ MPa m$^{-1}$, $n = 148$ vessels), but the two domains did not differ significantly in vessel radius. Indeed, most vessels of the outer xylem domain exhibited only minor flows (75% of outer vessels collectively contributed 0.1% of total flow), regardless of their diameter, indicating they were subject to very low axial pressure gradients. The most significant transverse pressure difference in our data was thus between the large vessels of the inner and outer domains, sustained by a band of small diameter vessels

and fibres that acts as a hydraulic barrier between successive domains of first-year xylem in *V. vinifera*[17,30].

The MRI map of water content (Fig. 2a) shows the outer vessels to be water-filled, which excludes embolism as a reason for their lack of function. Instead, a subset of these vessels appear to be hydraulically inactive due either to their own immaturity, as anticipated by Jacobsen and Pratt[31], or connection in series to non-conducting elements of the plant's hydraulic system[32]. There are also vessels in the outer ring that do carry significant flows (arrows in Fig. 2b), although they still appear subject to reduced gradients, when compared to the inner domain. This shows incomplete

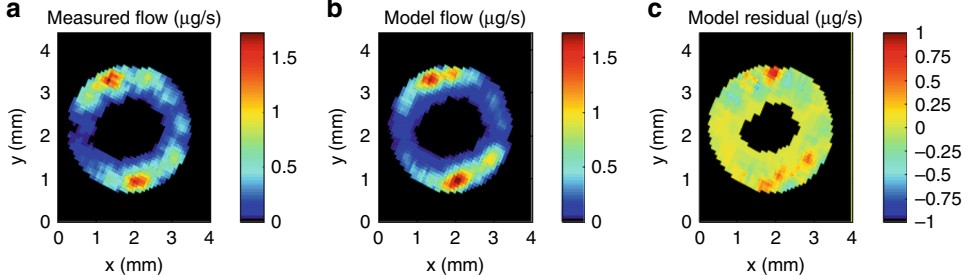

**Fig. 4 Masked scenario. a** Observed flow rates ($\mu g\,s^{-1}$, colour scale) averaged in 5-by-5 pixel window, cropped to the inner xylem domain (see Fig. 2c) only. **b** Model predicted flow rates ($\mu g\,s^{-1}$, colour scale) with single pressure set at top of domain, optimized over inner domain only, downsampled to 78 $\mu$m resolution and averaged over 5-by-5 pixel window. (**c**) Residuals between model and observed ($\mu g\,s^{-1}$, colour scale).

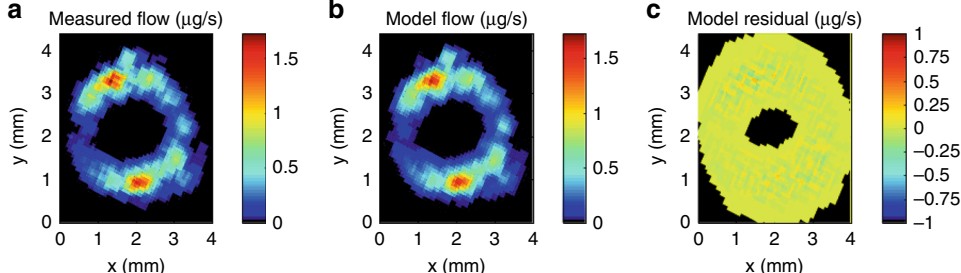

**Fig. 5 Independent pressures scenario. a** Observed flow rates ($\mu g\,s^{-1}$, colour scale) averaged in 5-by-5 pixel window. **b** Model predicted flow rates ($\mu g\,s^{-1}$, colour scale) with separate pressure value set for each vessel group ($n = 317$) at top of domain, downsampled to 78 $\mu$m resolution and averaged over 5-by-5 pixel window. **c** Residuals between model and observed ($\mu g\,s^{-1}$, colour scale).

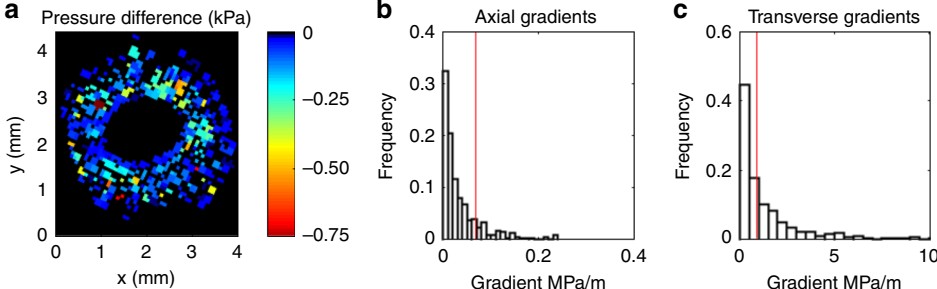

**Fig. 6 Axial and transverse pressure gradients. a** Pressure difference (kPa, colour scale) imposed on vessels between top and bottom of the stem in the best-fit model, downsampled to 78 $\mu$m resolution. **b** Histogram of resulting axial pressure gradients across the stem; red line indicates flow-weighted mean value. **c** Histogram of transverse pressure gradients across adjacent pixels in the top slice of the stem domain; red line indicates mean value.

hydraulic integration within the outer domain, as some, but not all, vessels are connected to transpiring leaves. The water-filled but poorly connected vessels are the main reason for a roughly threefold decrease in stem specific conductivity between the first (17.3 kg m$^{-1}$ MPa$^{-1}$ s$^{-1}$) and second (5.2 kg m$^{-1}$ MPa$^{-1}$ s$^{-1}$) scenarios. This result raises an issue for the method of measuring maximal stem hydraulic conductance[12], in which it is commonly assumed that all vessels capable of conduction in the excised stem also participate in flow in vivo, unless embolized. Instead, water-filled but hydraulically inactive vessels, which lack fully functional connections to active vessels may form a large proportion of total conduits, requiring greater attention to be paid to this factor in future measurements[9,33].

The diameter of vessels in the combined dorsal/ventral domains ($44 \pm 34$ $\mu$m, $n = 269$) is significantly greater than in the lateral domains ($30 \pm 20$ $\mu$m, $n = 222$, $P < 0.01$, $F = 33.97$, 2-way ANOVA), while the gradients are not significantly different across the two domains (i.e. dorsal/ventral vs. lateral). Although connectivity across the dorsal/ventral and lateral domains is reduced in grapevine xylem[22], the elevated share of

flow through dorsal and ventral vessels (78% flow in 55% vessels) was attributable to vessel diameter, unlike the elevated share of flow in the inner domain (91% flow in 70% vessels). By contrast, lower flow in the outer domain was due to vessel immaturity, immature vessel connections, or connection in series to other non-functioning hydraulic elements under development in more distal regions of the plant (e.g. leaves and differentiating xylem initials). Properties at both vessel scale (radius, length) and network scale (axial and transverse connectivity, placement of vessels in series or parallel with other resistances) vary across xylem domains and influence the hydrostatics of sap flow.

**Effects of network resistances.** Significant transverse heterogeneity in pressure gradients was evident even within just the inner domain of mature xylem that exhibited the highest flows. The following analysis is based on results from the inner domain alone, comparing the masked first scenario to the inner domain results in the second. Gradients in vessels that contributed significant flow ($>0.01$ $\mu g\,s^{-1}$) ranged from 0.01 to 0.45 MPa m$^{-1}$ (Fig. 7a). Transverse gradients at the top of the domain locally

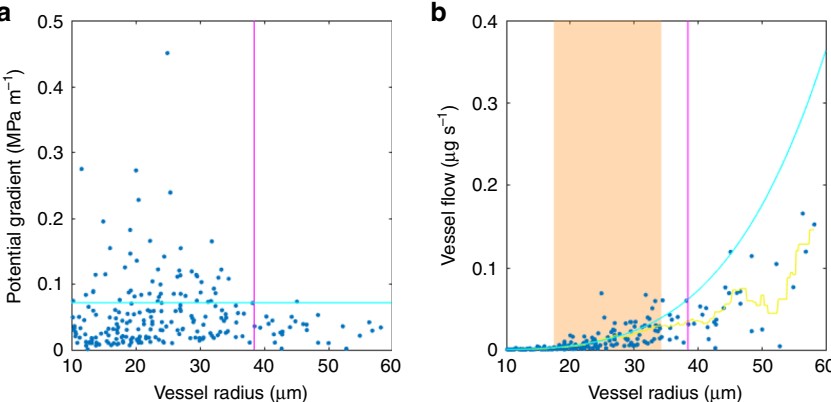

**Fig. 7 The relationship between vessel radius, pressure gradient and flow rate.** For all vessels in the inner region, with radius over 10 μm ($n = 306$), the gradient (**a**) and flow rate (**b**) are shown as a function of lumen radius. In a, the magenta line shows flow-weighted mean vessel radius and the cyan line shows flow-weighted mean gradient. In b, the magenta line shows flow-weighted mean vessel radius, the cyan curve shows idealized (Hagen–Poiseuille) flow rate over the range of radii 10–60 μm for the flow-weighted mean gradient, orange zone shows the range of radii over which the idealized relationship fits the flow-radius points (t-test of residuals from points in 6-μm window cannot reject 0 mean at $\alpha = 0.05$; p-values, t-statistics, and degree of freedom counts of separate tests in Supplementary Table 1), and yellow curve shows the running mean flow rate over a 6-μm window.

exceeded the axial gradient by roughly an order of magnitude. Even the vessel network of a single xylem domain, or of xylem not subdivided into domains, can thus exhibit significant heterogeneity of pressure gradients.

Such transverse gradients are the direct product of the structure of the angiosperm xylem network in general (both within and outside the scanned segment), with long, axially oriented lumina of variable but low resistivity separated by thin, transversely oriented high-resistivity pit membranes. If lumen and end-wall resistivity are roughly equal at the bulk scale[7], then the pressure drop across an average pit membrane connecting two vessels (0.185 μm thick for *V. vinifera*[34]), should on average be roughly equal to the pressure difference across the mean distance that water travels within one vessel; the average grapevine vessel is about 12 cm long[35]. It follows from the spatial organization of hydraulic resistances in the xylem network that transverse pressure gradients must locally exceed axial ones as the transverse resistance is concentrated in shorter intervals with greater resistivities.

The range of pressures implied by MRI observations at the top of the sample is the result of the spatially heterogeneous network of resistances beyond the μCT field of view (i.e. by end walls above and below the open lumen modelled here or by connection in series with less-conductive vessels). In a homogeneous network with uniform pressures imposed on either end (as in the first simulation), the gradient should be equal in all conduits. A non-homogeneous pattern of connection in a network of resistances (Fig. 8a) leads to pressure differences between vessels of identical length and width in a single cross-section. Moreover, increasing the resistance of a network element connected to a vessel lumen in series reduces the pressure drop across the lumen and results in lower flow through it (Fig. 8b). This is because more of the overall pressure drop needs to occur across the higher-resistance feature, leaving less for the lumen. The greater the difference in resistances between features connected in series, the more of the pressure drop will occur over the higher-resistance feature. As a result, such reductions in pressure drop are stronger for lower-resistance (wider) vessel lumina, as shown in Fig. 8c. We should thus expect both end-wall resistance and network effects to more strongly affect (reduce) flows in wider vessels.

In our second simulation scenario, focusing on the inner domain of fully functional xylem alone (and independent of the treatment of the outer domain), a trade-off between vessel radius

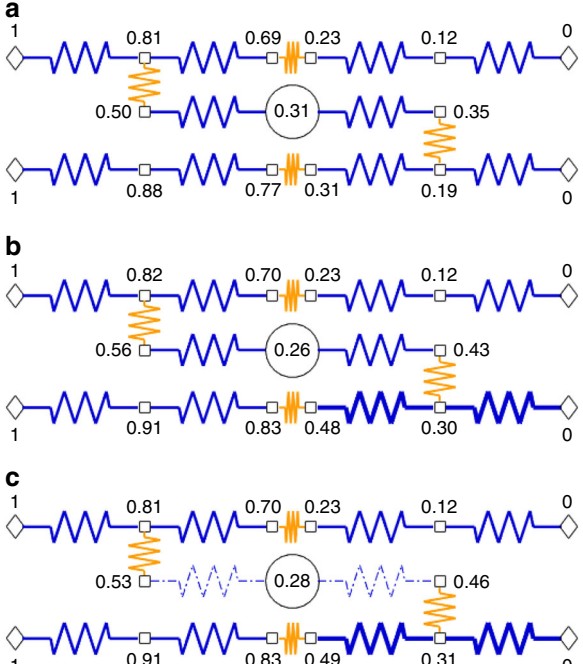

**Fig. 8 Ohm's law analogue diagrams of heterogeneously connected xylem.** Lumen resistances in blue, end- and side-wall resistances in orange. Pressures set at end nodes (diamonds) to 1 normalized unit upstream and 0 downstream and calculated at other nodes (squares) from network of resistances; flow rate through central vessel in normalized units also shown (circle). **a** Each lumen resistance shown has a value of 0.25 normalized units and wall resistances set to 1. Pattern of connections results in pressure differences among vessels in any given transverse (vertical) slice. **b** Lumen resistance in bottom-right vessel doubled, pressure drop and flow rate across central vessel decrease. **c** Lumen resistance in central vessel halved, pressure drop decreases further so that flow rate only partially recovers; note that, due to network position, flow rates in the central vessel (0.28 units), the top-right vessel (0.46, flow rate not shown) and the downstream lumen element of the bottom-right vessel (0.62, flow rate not shown) are ordered inversely to their resistivities (assuming all lumen elements of equal length), i.e. modelled width.

and pressure gradient emerges. The flow-weighted mean vessel ($n = 343$) radius was $38.5 \pm 11.4\,\mu m$ (weighted mean ± standard deviation found from the weighted variance with reliability weights) and the flow-weighted mean axial gradient $0.071 \pm 0.071\,MPa\,m^{-1}$. All vessels but one were below this level in either radius or gradient (Fig. 7a), indicating an important trade-off between the two variables in our results. Indeed, vessels with a radius above $38.5\,\mu m$ ($n = 27$) have a significantly lower mean pressure gradient than vessels with a radius below $38.5\,\mu m$ ($n = 316$, $F = 4.58$, $P = 0.033$, 2-way ANOVA). Using the mean pressure gradient, we found the curve of predicted HP flow rates spanning the range of radii $10–60\,\mu m$, which contained all vessels contributing significant flow in vivo (Fig. 7b). Flows in vessels ranging from $17.5$ to $34.3\,\mu m$ in radius followed this theoretical prediction closely (residuals distributed with mean 0, see Methods for detail and Supplementary Table 1 for results of individual t-tests). In vessels with a radius above $35\,\mu m$, flow rates deviate significantly ($P \ll 0.01$; results of individual t-tests in Supplementary Table 1) from the relationship due to lower than expected pressure gradients (Fig. 7a, Supplementary Fig. 2a). Our results thus indicate a statistically and physiologically significant negative relationship between vessel radius and gradient, such that the increase in flow between smaller and larger vessels is less than would be predicted from the increase in radius alone due to a systematically lower pressure gradient in the largest vessels (Fig. 7, Supplementary Fig. 2).

This radius-gradient trade-off had significant effects at the vessel and tissue scales. Compared with the masked simulation in the first scenario, vessels of above-average diameter lost 23% of their flow to narrower vessels in the second scenario. This effect increased with vessel diameter, for vessels of above-average diameter (Fig. 7, also linear regression of flow difference between simulations on diameter for all vessels above mean diameter, slope $870 \pm 150\,\mu g\,s^{-1}\,\mu m^{-1}$, mean ± SE, $F = 33.8$, $P \ll 0.01$, $R^2 = 0.18$, $n = 27$). At bulk tissue scale 15% of overall flow was redirected from wide to narrow vessels, as compared to predictions based on the simple application of HP and the homogeneous xylem model.

As shown in Fig. 8, such relative decreases in flows through wider vessels result from a disproportionate effect of network heterogeneity on the pressure drop across these conduits. Both connection in series with narrower vessels and connection through end-wall pit membranes of equal total resistance (leading to a greater ratio of wall to total vessel resistivity, or 'wall fraction', than in narrower vessels) would have such an effect. It may be that vessels wider than average are simply inherently likely to be connected to vessels with narrower lumina than their own. On the other hand, it is noteworthy that the total specific conductivity of the inner region in the second simulation ($7.85\,kg\,m^{-1}\,MPa^{-1}\,s^{-1}$) was about 41% lower than in the first ($13.4\,kg\,m^{-1}\,MPa^{-1}\,s^{-1}$), which effectively assumes the medium to be homogeneous. This conductivity drop corresponds remarkably well with observations of bulk wall fraction in grapevine (38%)[7], as well as the theoretical optimum wall fraction (40%)[11]. This suggests the dominant source of axial pressure gradient heterogeneity in our results may be the resistance of vessel end walls. No vessel had end walls within the sample imaged with μCT and their effects on observed in vivo flow were implicit in the pressures inferred from the modelled vessel network. Whether the observed flow reductions in wider vessels are due to higher wall fractions or to connection in series with vessels of diameters closer to the mean remains an open question.

**Dynamic aspects of xylem hydraulics.** Pressure gradient heterogeneity can be influenced by the hydrodynamics of xylem flow

in ways not considered by our model or not captured in our observations. As the plant dries and xylem pressures become increasingly negative, increasing numbers of conduits will embolize, meaning they will be blocked for water flow by air. If certain vessels are blocked off, this clearly changes the overall network structure of resistances and will result in new, dynamically changing patterns of water flow. While MRI revealed no embolism of wide vessels in the plant, we cannot fully exclude the presence of embolism in vessels outside the scanned region.

We nevertheless believe our results to reflect the unstressed state of the xylem network, rather than of embolism. The plant was well watered throughout the experiment and grapevine xylem has repeatedly been shown to exhibit minimal embolism when plants are maintained in a well-watered state[36–38], or even at wilting point[39]. Experience from these studies also shows embolism does not arise in grapevine due to handling using the protocols of the present study, including during insertion into the MRI instrument.

Moreover, while certain species are known to experience embolism in wider vessels first[40], in grapevine embolism has been shown to spread radially out from the pith instead and shows no relation with vessel diameter (see Supplementary Fig. 2 in Brodersen et al.[36]). We thus find no reason to believe that incipient embolism would lead to a redirection of flows away from the widest vessels in grapevine, as was found here.

As shown in Fig. 8, transverse pressure gradients can arise in a hydrostatic network model, with uniform pressures on the boundaries, due to the heterogeneity of the medium alone. As vessels are hydraulically connected to storage elements within the xylem[41], these may serve to amplify or dampen axial and transverse pressure gradients, depending on their distribution and system history. A hydrodynamic model, with an explicit storage component, may provide additional opportunity for such transverse gradients to arise, under certain conditions, and may in principle better capture the detail of micro-scale xylem hydraulics. Indeed, hydraulic capacitance can be an important element of the water relations of large trees[42,43]. The magnitude of storage pools in the xylem, however, is negligible compared to the flow rate during transpiration in many plants[44], including first-year grapevine stems and thus is unlikely to shape pressure gradients within our model domain to any significant extent. Moreover, it has been shown that even storage pools theoretically available in the xylem can remain largely unused in vivo[45,46], meaning that their effect on local pressure gradients is likely less pronounced than their total capacity suggests. Finally, while significant release of water from storage can temporarily affect transverse pressure gradient patterns, it is unclear why this mechanism should lead to a decreased overall contribution of widest vessels to flow. For all these reasons, we do not believe it is a good candidate mechanism to explain our result.

**Discussion**

Pressure gradients within the xylem result from a combination of properties of the medium and the pressures imposed at its boundaries (by leaves, roots). If we consider that each leaf along the stem imposes a particular pressure value at a point along the xylem perimeter, the gradients that develop in the xylem will depend on pressures imposed and how the medium propagates them. Xylem anisotropy, specifically increased transverse resistivity, is a necessary condition for the emergence of strong transverse gradients. If transverse resistivity were much lower than axial, pressures would be about the same in each cross-section and would change linearly along the stem, between subsequent leaves. With a high transverse resistivity, non-uniform boundary conditions and xylem heterogeneity are each separately

sufficient to achieve transverse gradients. To virtually eliminate transverse gradients in a homogeneously anisotropic xylem, the pressures set by leaves would have to align to a linear axial gradient. Any departure by a given set point would yield transverse gradients in the surrounding xylem, as a function of the ratio of axial to transverse resistivity. Similarly, though, even with pressure set points that align perfectly to a single linear gradient along the stem, any departure from xylem homogeneity will also generate transverse gradients and redirect flows, as illustrated in Fig. 8.

The stronger the heterogeneity is, the greater the transverse gradients become, and the greater, also, is its effect on flows and overall xylem conductance. For example, to determine the theoretical conductance of the network in Fig. 8 based on information about lumen resistances from halfway to three-quarters along the network (i.e. a network segment that includes half the lumina of the central and two right-hand vessels) we sum the lumen conductances for these segments and divide by four to scale to the entire network. Comparing these values to the 'observed' ones (i.e. flow divided by pressure difference) yields a ratio of 0.41, 0.43, and 0.31 (observed:theoretical) in panels a, b, and c respectively. This value, bulk conductance of the vessels divided by the conductance of observed lumina alone, is broadly analogous to one minus the "wall fraction" calculated by previous authors[7,11]. Yet we achieve change in this value not by changing end-wall resistances, but lumen diameters. This may actually modulate the effective wall fraction, since end-wall resistance has a greater effect on wider vessels, but we cannot disentangle this from the effect of lumina of different diameters being connected in series. In sum, it is the effect of a changing level of heterogeneity in the network.

In the foregoing thought experiment, we have also not changed the pattern of network connections (topology) between panels. Previous modelling studies have found that conduit connectivity has a strong effect on network "efficiency" and thus on the hydraulic conductance of such theoretical vessel networks, which cannot be considered merely as a sum of their conduits[15,16]. This study points to a separate effect, where the absolute differences in resistances of connected elements, rather than conduit topology, contributes to the heterogeneity of the medium and thus affects both local and bulk flows. The vessel flow rates inferred from our in vivo observations are the first indication that such effects play an important role in the living plant and disproportionally reduce flows in the widest vessels below HP predictions.

Our results point to important implications of the xylem network's heterogeneous organization that affect the functioning of network constituent parts and vice versa. Bottom-up effects include the greater propagation of gentle axial gradients along the stem by vessels of greater length and diameter, which thus contribute disproportionately to the emergence of transverse pressure gradient heterogeneity. The presence of inter-vessel connections along the length of vessels rather than exclusively at their ends means that the average distance that water travels within a single conduit is less than the physical vessel length. Known top-down effects include xylem sectoriality, or disconnection across xylem domains, so that multiple parallel networks can effectively sustain multiple flow regimes in the same stem segment. But even within a domain of connected xylem, the arrangement of resistances at the network scale redistributes pressure gradients towards the highest resistances. Our study suggests that at least in certain types of xylem, this will increase flow in the smaller diameter vessels and reduce it in the widest vessels. Such interactions across scales show that questions about optimal vessel dimensions[8] or the relations between end-wall resistance and vessel size[10], cannot be answered meaningfully for individual vessels but only with due consideration of the medium as a whole.

Recognizing the importance of xylem heterogeneity puts some old questions in plant hydraulics, concerning the construction of xylem, in a new light. To the extent that the redirection of flow found here arises mainly due to vessels of unequal diameter connected in series, we should expect that the contribution of wider vessels to flow depends on how far from the mean diameter they deviate or how well connected they are to other wide vessels. We would thus expect the emergence of two separate strategies of xylem construction: one in which vessel diameters cluster narrowly around the mean (as in diffuse porous woody species) and another with preferential connections among vessels of similar diameters (as in ring porous species). The former strategy essentially tries to limit the degree of heterogeneity in the network, whereas the latter seeks to overcome its effects by adopting a particular organization of the heterogeneous medium at a larger scale, effectively establishing preferential flow paths. That we found a particularly strong redirection of flows while investigating the first-year stem of a vine suggests that it exhibits neither strategy. Indeed, it is known that vessel diameters in grapevine stems taper acropetally[47] and are somewhat constricted at nodes, which likely contributes to the strength of this effect in our data. If heterogeneity arises mainly by connecting vessels of dissimilar diameter in series, then significant redirections of flow may not be found to generalize beyond vines, to other woody plant functional types. Nevertheless, the principles behind the effect clearly constrain the xylem construction of all plants.

To the extent that flow redirection instead results mainly from a greater effect of end-wall resistance on wider vessels, we would expect the effect to be a significant driver of the safety-efficiency trade-off. Increasing lumen diameter without increased pit area of the vessel walls results in decreased pressure gradients in the lumen and limited increases of total vessel conductance[7]. On the other hand, according to the rare pit hypothesis, increasing pit area in proportion to the increase lumen conductance should also increase vessel vulnerability to embolism spread[11]. In this case, we would also expect the effect to generalize well beyond vines, as it would be driven by the lumen and wall resistance ratio of each vessel.

As the two mechanisms are not mutually exclusive, we would expect both to constrain the overall parameter space, in which xylem construction can be optimized. Together, the two may imply that for wide vessels to truly confer a conductivity advantage, they must always be organized into preferential flow paths with a high degree of connectivity between each pair of elements and thus also a disproportionate vulnerability, following the rare pit hypothesis. Understanding such implications of xylem heterogeneity moves us closer to unravelling the causal links between its features at different scales of organization and thus represents a significant step towards a comprehensive porous medium description of the xylem.

## Methods

**Plant material**. Grapevine (*Vitis vinifera* L., "Cabernet Sauvignon") grafted on SO4 rootstocks were planted in 10-L pots with commercial potting soil and grown in a greenhouse (16 h light/8 h dark, 22 °C/24 °C, RH 50/70%) under ambient light. Artificial light ($\sim$200 µmol m$^{-2}$ s$^{-1}$, Philips SON-T Agro 400 W, Philips, Osnabrück, Germany) was added whenever ambient light was <390 µmol m$^{-2}$ s$^{-1}$. The plant was well-watered throughout its growth. The vines were pruned to yield a single stem without side branches, reaching a length of 2.5 m. After gently placing it in the MRI the plant was allowed to recover overnight. MRI velocimetry was done the following day under continuous light (300 µmol m$^{-2}$ s$^{-1}$, 21 °C, 60% RH).

**Imaging**. All MRI was done at the IBG-2 Plant Sciences institute at the Forschungszentrum Jülich, Germany. The MRI scanner comprised a Varian spectrometer, 4KW RF amplifier, a set of 400 A gradient amplifiers, and a 800 mT m$^{-1}$ biplanar gradient set (Tesla engineering, Storrington, UK) mounted in a 1.5T split wide bore superconducting MRI magnet (Magnex Scientific, Oxford, UK), adapted for tall plants. For excitation and signal reception a ø 15 mm, 10-turn solenoidal RF

coil was wound on the middle of the second internode from the graft, before placing the plant in the scanner and connecting the coil to its tuning mechanism.

MRI flowmetry was done using a Pulsed Field Gradient—Stimulated Echo—Multi Spin Echo (PFG-SE-MSE) sequence[48] with experimental parameters: field of view (FOV) $10 \times 10$ mm, slice thickness 3 mm, matrix size $128 \times 128$ pixels, repetition time ($T_r$) 2.5 s, no averaging, spectral width (SW) 50KHz, $T_E$ 4.7 ms. Flow encoding was done stepping the amplitude ($G$) of the PFGs from $-G_{max}$ to $+G_{max}$, sampling q-space completely and equidistantly in 64 steps, using a flow labelling time ($\Delta$) of 20 ms, a PFG duration ($\delta$) of 3 ms and a $G_{max}$ of 0.5 T m$^{-1}$, in this sample allowing the detection of xylem sap flow velocities of up to 5 mm s$^{-1}$. Only the first echo was used. A quantitative map of volume flow (Fig. 2b) was obtained by analysing singular xylem flow measurements on a per-pixel basis[49].

Water content maps were acquired by means of a CPMG type sequence[50] with settings: FOV $10 \times 10$ mm, slice thickness 3 mm, matrix size $256 \times 256$, number of averages 2, echo time: 6.3 ms; number of echoes: 64; $T_r$: 2.5 s; SW 50 kHz. The acquired MRI datasets were fitted on a per pixel basis using a mono exponential decay function, yielding a quantitative map of amplitude (Fig. 2a).

Following MRI imaging, the stem segment used for the flow measurements was excised and stored in a 1:1 deionized water:ethanol solution for shipping to the Lawrence Berkeley National Laboratory in Berkeley, CA, USA. The stem segment was allowed to dry under ambient conditions to remove water from the vessel lumina prior to imaging with the μCT system (beamline 8.3.2) at the Advanced Light Source following the methods of Brodersen et al.[30] The stem segment ~2 cm long was mounted in the sample holder and imaged at 24 keV with an exposure time of 300 ms as the sample rotated around 180˚, resulting in 1024 raw projection images. The 3D dataset was reconstructed using a custom Python script to produce a final image stack with a final voxel resolution of 3.2 μm (cross-section in Fig. 2c). We then used the TANAX and INFLOW software programs developed by Lee et al.[17], Brodersen, et al.[30] to digitally extract and characterize the complete vessel network from the segment, and then model the network performance as described below.

Spatial coordination between datasets was achieved manually in a way that maximized overlap in phloem rays in both sets of images (Fig. 2d). Algorithmic maximization of image overlap did not yield increased fit and the observed minor mismatch between vessel positions in the two datasets is taken to be inherent to the method. It is likely mostly due to the scale mismatch between the two imaging technologies: μCT produced 3 mm of individual 3.2 μm slices, while MRI imagery integrated over the 3 mm stem segment. Other possible sources of mismatch were xylem deformation due to ethanol preservation in transit and relative sensor placement despite stem marking to ensure scans of the same region. Xylem cross-sectional area was determined by counting classified xylem pixels (inside cambium and outside pith) in μCT images.

**Flow modelling**. We used segmented μCT images to define a xylem network domain following Brodersen et al.[30] The distribution of pressures and flows on the resulting domain was calculated using custom software[17], which sets the resistance of vessel lumina using the HP equation and uses inter-vessel pitting area along with the area-specific pit resistance value for grapevine of 168 MPa m$^{-1}$ s$^{-1}$ for connections[11], whose distribution is inferred from the segmented μCT scan. A full description of the flow model is available in Lee et al.[17]

We defined an objective function that is minimized as the modelled flow rates approach those observed with MRI. The function sums flow rates of individual vessels within the pixels of the MRI flow data and finds residuals between this downsampled model flow and the observations. To account for tissue deformation in transport between MRI and μCT measurements, it averages residuals over a 5-by-5 pixel neighbourhood before finding the sum of their squares. The objective function was minimized by allowing the pressures at the top of vessels to vary and seeking the best-fit value with the trust-region-reflective algorithm[51,52].

**Analysis and statistical methods**. We report a number of values (e.g. pressures, gradients, radii) as flow-weighted mean, across a region of the xylem (MRI pixels, a particular xylem domain, or the xylem as a whole). This means we took a weighted average of the respective quantities (for individual vessels or μCT voxels, as noted in the text), using corresponding values of flow as the weights.

A flow-weighted mean is the most appropriate averaging method if the contribution of each vessel or pixel to the overall flow is being considered. All mean values are reported together with standard deviations. For weighted means, standard deviations are derived from the weighted variance with reliability weights.

We evaluated model fit using NSE, a measure common in hydrological sciences[27], ranging from $-\infty$ to 1, where a value of 0 indicates the model is only as good at predicting the observations as is their mean value and 1 indicates the model matches the observations perfectly. We found specific conductivity for a given domain by summing flow from all its vessels and dividing by the flow-weighted mean pressure difference across them and the domain area.

We performed a two-way ANOVA to determine significant differences between gradients and radii across radial (inner vs. outer) and tangential (dorsal or ventral vs. lateral) domains. All significant differences are noted in the text; interactions between radial and tangential domains were not significant for either variable. Differences between gradients of narrow and wide vessels in the inner xylem domain were determined by one-way ANOVA.

To evaluate the HP flow rate predictions, we found the residuals between the vessel flows from the best-fit model output and the theoretical flow rate and then used a sliding 6 μm window to perform a two-tailed t-test of the hypothesis that the residual distribution has a mean of 0. Results did not differ significantly for window widths up to 12 μm, as in all cases p-values crossed the 5% significance level between 34 and 35 μm of radius and dropped below $p = 0.001$ around 35 μm. They stayed at $p < 0.001$ over the remainder of the domain (radius > 35 μm) for wider windows and $p < 0.05$ for the narrowest ones, which led to a short interval with few data points, increasing uncertainty locally.

All computation was done in MATLAB (MathWorks, 2017).

**Reporting summary**. Further information on research design is available in the Nature Research Reporting Summary linked to this article.

## Data availability
All calculation inputs (including raw MRI images) and outputs, including all data reported in figures here are available online at https://bitbucket.org/xylemlab/invivoflow/. The reconstructed 3D μCT scan is available at Open Science Framework repository with https://doi.org/10.17605/OSF.IO/EY2TJ.

## Code availability
All original code used in this work is available online at https://bitbucket.org/xylemlab/invivoflow/.

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

## Acknowledgements

The Advanced Light Source is supported by the Director, Office of Science, Office of Basic Energy Sciences, of the US Department of Energy under Contract no. DE-AC02–05CH11231. This research was partially supported by the USDA Sustainable Vineyard Production Systems CRIS Project No: 2032-21220-006-00D. We thank the IBG-2: Plant Sciences institute at the Forschungszentrum Jülich for providing MRI measurement time and dr. Dagmar van Dusschoten for programming the MRI sequences.

## Author contributions

C.W. performed MRI image acquisition and analysis. A.M. and C.B. performed μCT imaging and contributed to model development. M.B. performed μCT image analysis and flow modelling. All authors contributed to research design and manuscript composition.

## Competing interests

The authors declare no competing interests.
