## [Peer Review File · Nature Communications]

Reviewers' comments:

Reviewer #1 (Remarks to the Author):

This paper uses an ingenious approach based on MRI, microCT, and model-fitting to document flow and pressure gradients in a xylem vessel network, resulting in an entirely novel conclusion that narrower vessels contribute more to sap flow than previously predicted. The measurement design was excellent, analyses appropriate, and the conclusions are strongly supported by the data. I expect that some fellow reviewers might complain that the experiment was done with a single grapevine stem, but that single stem yielded a wealth of data and well-supported analyses, so I do not see what replication would have added to this study. Certainly, future research along these lines should follow up by comparing different varieties and species, but it would not be fair to demand such comparisons from this first study.

I have only one thing to suggest for improving this most important aspect of the manuscript, the redirection of flow to narrower vessels: It might be helpful to readers to add some future research questions in the discussion. For example, what might these findings imply for the effect of embolism in the functioning of the vessel network? What do the findings imply for different degrees of connectivity, different vessel diameter distributions, and different hydraulic integration found in different species? The authors undersell their findings a bit in the discussion by not clearly describing a vision for how their research will affect the field of plant hydraulics. The conclusion that xylem networks are more than the sum of their parts has been reached many times before, so the last sentence of this paper does not do full justice to the counterintuitive and novel findings of this study. The discussion deserves a stronger elevator pitch at the end to make a case for how the findings might affect future thinking in the field.

Another important contribution of this study is the conclusive evidence that not all filled vessels are hydraulically active, at least not in this grapevine variety. This is not a new observation, but it contradicts a long-standing assumption in the field of plant hydraulics that non-conductive vessels are mostly embolized and implies that xylem vulnerability curves created by microCT must be interpreted with caution, as not all filled vessels can be assumed to be functional. The authors do not refer to Choat et al. (2010. Measurement of vulnerability to water-stress induced cavitation in grapevine: a comparison of four techniques applied to a long-veesled species. *Plant, Cell & Environment* 33:1502-1512), which was coauthored by McElrone and Brodersen and in which vessels being filled was equated with vessels being functional for water transport. That paper created strong responses in the plant hydraulics community, and none of these responses are cited either, except for Jacobsen et al. (2018). I do not disagree with the decision to leave out much of that discussion about impacts of long vessels on vulnerability curves, because that is not what this paper is about. However, I think it would be fair and appropriate for the authors to concede that Jacobsen and Pratt (2012. No evidence for an open vessel effect in centrifuge-based vulnerability curves of a long-veesled liana (*Vitis vinifera*). *New Phytologist* 194:982-990) were correct when they suggested that many vessels in grapevine are non-functional even when filled and that Choat et al. (2010) failed to take that into account. This should be a clear warning to others who interpret microCT images in terms of xylem vulnerability to embolism. Of course, this has very little bearing on the question whether long vessels create artifacts in hydraulic measurements, and there is no reason to address that question in this paper.

I also suggest that the recent paper by Pratt and Jacobsen (2018. Identifying which conduits are moving water in woody plants: a new HRCT-based method. *Tree Physiology* 38:1200-1212) should be cited, because it addresses the same general research question as in this study, also using grapevine and microCT. However, it is fair to say that the combination of MRI, microCT, and modeling turned out to be far more powerful than the microCT/staining approach taken by Pratt and Jacobsen (2018).

Overall, I was very impressed by this study and enjoyed reading the manuscript and thinking

about its implications for water transport in plants. The manuscript is in excellent shape, and beyond the couple of suggestions above the only other comment I have is about a typo in line 542, where "fig. 1D" should read "fig. 2D".

Jochen Schenk, California State University Fullerton

Reviewer #2 (Remarks to the Author):

In this manuscript the authors present results of MRI (volume) flow rate measurements, X-ray microCT and simulations of xylem anatomy and flow in grapevine. The authors conclude that the MRI results (observed in an intact plant) can only be simulated (on the basis of the anatomy of a 3.3 mm plane) if transverse pressure differences are applied in the simulation. This is due to the vessel network heterogeneity. These transverse pressure gradients redirect flow from wide to narrow vessels, reducing the contribution of the wider vessels to overall sap flow. The relation between xylem anatomy, flow, efficiency and vulnerability of xylem to cavitation is of wide spread interest and is discussed in a large amount of papers. The presented results, if reliable, certainly have important consequences in answering questions about optimal vessel dimensions, xylem conductivity and modelling (dynamics) of flow through plants. So as such the manuscript covers a topic of high interest for a broad audience and is certainly of relevance.

The conclusion is quite contra-intuitive. There is quite some literature available trying to correlate cavitation resistance and hydraulic conductivity and the very complex tissue, vessel network and pit characteristics. The conclusion in this manuscript is based on a combination of three input parameters: (1) volume flow rate (MRI), (2) xylem anatomy of a 3.3 mm slice (microCT) and (3) modelling.

(1) In the manuscript the reliability of the MRI flowmetry cannot really be evaluated. No test results or verification of the MRI results are presented. And only one (?) plant has been measured (true?? I cannot verify from the presented info). MRI on (single) vessel flow has been presented before and has already shown the existence of differences in (local) pressure in individual xylem vessels within the plane imaged (Scheenen et al, Plant Physiol 2010). In that paper also the MRI parameter settings in relation to the dynamic range for the flow measurements (maximal velocity!) have been discussed. Unfortunately, it is impossible on the basis of the presented information in the manuscript to verify if the MRI measurements have been done in the correct way: data on maximal velocities and on small and big delta values of the flow encoding magnetic field gradient are missing. If the dynamic range of the MRI flowmetry was not properly set it can easily explain why larger vessels/high flow rates (which are expected to have the largest velocities) are under estimated (Scheenen et al, 2010).

A second question related to the MRI experiment is the presence of cavitations. Plant handling to insert the plant in the MRI set up can easily induce cavitations. Xylem in grapevine is quite vulnerable to cavitation, also due to handling/mechanical damage (Jacobsen et al., 2005). Normally it lasts a couple of days to restore flow (if all!) to the maximal values under the given conditions in the MRI system (see e.g. Scheenen et al, 2010, three days for a cucumber plant; Windt et al, 2006). The fact that the MRI water density map (amount of water per volume, not water content!, line 196) shows that the vessels are water-filled does not guaranty that in the xylem system no cavitations are present. Only a slice of 3 mm is imaged by MRI, but the xylem extends over much larger distances. As demonstrated in Scheenen et al, 2010, vessels can be present that are filled but with zero flow, even very close (within the same vascular bundle, so most probably hydraulically connected) to fully functioning vessels. Refilling vessels show a low velocity (see also Kaufmann et al, . If we consider such situation isolated we indeed have to conclude that local transversal differences in pressure may exist, but the reason is given somewhere else in the xylem system!

Large volume vessels are most vulnerable to embolism (see among many others the recent paper of Jacobsen et al, IAWA Journal 40, 2019). Large diameter vessels are also the longest ones. Again, embolism in such vessels can easily explain why flow in the larger vessels (and also why

the outer xylem domain) are underestimated, resulting in the presented transverse heterogeneity in pressure differences. Embolism in part of the xylem can even explain strongly reduced flow in xylem regions (Jacobsen et al, 2019; Mrad et al, 2018).

(2) The anatomy was measured in a segment of 3.3 mm by microCT. As has been demonstrated in many papers the hydraulics and flow patterns (and pressure drops) are the result of the total xylem anatomy. Jacobsen et al, 2019, has recently suggested that vessel network traits like vessel diameter and length, number of pits (which tend to be non-evenly distributed over short and long vessels), bordered plates, etc, may be important in developing methods for microCT analysis that are able to accurately predict xylem function. MRI has been measured in a 3 mm slice (same position as the microCT analysed segment) within the intact plant system. Flow (and pressure drops) is the result of the whole network. E.g. embolism and other resistances upstream or downstream the slice will affect the flow observations. I wonder if the MRI measured flow pattern in the slice can be modelled by the anatomy of that slice only! Does it correctly represent the xylem network for modelling the (MRI) observed flow?

In normal practice of hydraulic conductivity and flow measurements, pressure drop and anatomy are related to one and the same segment. In such a setup, flow in a stem segment of grapevine was measured by MRI (Windt, PhD thesis, p 58:). In that situation the largest vessels contributed (by far) the most, only 31 % of available xylem area was used, and I cannot find any indication of redirection of flow from wide to narrow vessels.

Some minor questions about the microCT results: 199 solitary vessels, 189 vessels with a single connection, and 103 with more than one (lines 18-139) were reported. What type of connections were included? Pits and/or relays? What resistance values have been used? How were the connections distributed? (see next remark in relation to fig 8).

(3) Modelling the dynamics of water flow through plants and trees has attracted continued interest for many years (Tyree, *Tree Physiol* 1988; De Schepper and Steppe, *J Exp Bot* 2010; Zweifel et al, *J Exp Bot* 2007). Such models may be divided into two major classes depending on whether they account for the water storage capacity of stems and leaves, i.e., whether they are dynamic or steady state models, respectively. Water stored in the stems may provide a substantial amount to daily sap flow, but this water is under well-watered conditions replaced by inflows at night (see e.g. review by Landsberg and Waring, *Tree Physiol* 2017). These observations show the importance of storage pools that are connected to the xylem. This has been included in many (hydraulic) models that describe whole plant water (and carbon) flow (see e.g. De Schepper and Steppe, *J Exp Bot* 2010; Zweifel et al, *J Exp Bot* 2007).

Xylem vessels cannot be considered as isolated from the storage pools. See also Edwards et al, *Tree Physiol* 1986, who present relation between water potential and relative water content, and relation between relative water conductivity and relative water content. As long as transverse pressure differences exist there is a non-equilibrium (pressure, and water potential) situation and radial transport will occur and storage pools cannot be ignored! To a certain amount there will be redistribution within xylem (depending on elasticity), storage pools/cells in different tissues. Such storage pools are not included in the presented model to explain the experimental results, but will certainly affect the outcome and conclusions!

In the second scenario to model flows pressure at the top of the sample were set independently in each of the 317 vessel groups (lines 162 – 163). It is not clear to me if at the base the pressure was set equal for all vessel groups. If so, why?? If on the top of the slice (in vivo!) transverse pressure gradients are allowed, why not at the base?? What is essential is the (axial) pressure difference or pressure gradient that drives flow. If transverse heterogeneous pressure gradients exist over the inspected slice they will exist over the whole length of xylem network and the details of the slice only will not be sufficient to describe flow in the whole system, as argued above. What are the consequences for the conclusions?

The Ohm's law analogues presented in fig 8 (and discussed in section Effects of network heterogeneity, lines 246-260 and 288-303) is somehow suggestive. If more (or/and earlier to the left) connections are introduced the outcome will be quite different!

In summary:

- a very interesting manuscript, but essential information is missing (MRI experiment), it is not

clear if flow can be modelled on the basis of the anatomy of a 3 mm slice, and it is not clear if the presented model (including scenario 2) is correct under the conditions of transverse pressure gradients.

Reviewer #3 (Remarks to the Author):

Review of In-vivo flow redirection from wide to narrow xylem vessels due to pressure gradient heterogeneity

Summary

This work reveals the different in-vivo effects that lead to a deviation of bulk xylem hydraulics from the widely used simplistic addition of Hagen-Poiseuille flows. To do that, this study uses state of the art imaging techniques coupled with a recently developed software to computationally resolve the xylem network. This is then fed to a software that computes water flow through the resolved network.

The premise of this paper is interesting and important to the field. The authors make a compelling case of why their study is needed: in the literature, there are no complete theoretical treatments that upscale in-vivo plant vessel, pit membrane, etc. hydraulic functioning to bulk organ water flow. The introduction primes the reader about the goal of the work and sets it up against other work in the field, thus providing a good argument for its originality. Another quality that must be mentioned is the ease of readability of all the figures. The results of this paper, although not surprising, do provide crucial and previously not available information for scientists to advance the understanding of plant functioning in-vivo. Moreover, the authors did a good job discussing what their analysis of a small segment can and cannot say by enumerating possible external (outside the analyzed domain) factors that might have led to the observed results. These observations will help achieve the much-needed porous medium description of plant xylem in the future.

Major concern

My only major concern is that the observed trends seen in this specific xylem segment might be different or reduced in intensity in different segments and in different angiosperm plant types. The authors do point out that a large portion of the transverse pressure gradients happens because, for example, large vessels are more likely to be connected to a smaller one downstream of the observed segment. If this is true, might we see a less pronounced effect of transverse pressure gradients in, for example, self-supporting ring-porous angiosperms where large vessels in the earlywood are connected to other large vessels? To sum up, 1) how general are the observations seen in this segment? 2) To what extent is this effect particular to vines?

Addressing this point is especially important because neither the title nor the abstract mention that these experiments are done on a vine.

Line by line comments and minor concerns

1. Line 51: "Volume flow rate" instead of just "flow" so some readers don't confuse this with velocity. Velocity varies with R to the 2nd power. But the volume flow rate varies with R to the 4th power as stated.
2. Line 51: "such that a small increase" instead of "and a small increase".
3. Line 61: Say "a bulk form of the HP relation..." because even though each vessel might follow the HP relation if we knew the pressure gradient along each of them, the whole vessel network might not conform to the addition of all HP flows from the individual vessels.
4. Line 76: "approximate" and not "determine"
5. Line 121: "Figure 2B"?

6. Line 122: "Figure 2A"?

7. Lines 142-150: One shouldn't really expect that the pressures on top of the segment to be the same along the whole cross-section. This is because the observations are in-vivo so the boundary pressures are set at another level not captured in the study segment (the leaves) as explained by the authors later. The value of that simulation, however, to me, is to show the extent of how external factors might (outside the analyzed domain) contribute to the development of the large transverse pressure gradients. So, it is worth casting the first simulation not as an attempt to model the flow correctly, but as a comparative experiment for the second, more physically accurate, simulation.

8. Lines 243-244: In line 138, you say "... 199 are solitary vessels" and this is out of only 491 total vessels in your segment. About 41%. So my gut feeling is that transverse resistances through pit membranes aren't the only story here. It is also about vessel connection patterns upstream and downstream your resolved segment.

9. Lines 267-268: you repeat the word "gradient" twice (read as the gradient of the gradient), fix this sentence, please.

10. Line 343-344: I wouldn't say that heterogeneity "rather than" topology affects both local and bulk flows. Vessel geometry and topology throughout the whole plant in fact lead to pressure gradient heterogeneity. These aren't competing effects, but rather one results from the other although not solely.

11. Figure 1 caption: Have the vessels been artificially separated during the generation of graphics? How come there are connections where vessels do not touch physically?

Reviewer #4 (Remarks to the Author):

General comments

An update to working models might be useful as the authors point out, but it is also worth noting that predictions of flow scaling with the fourth power of conduit radius neglect to consider effects on flow of perforation plates or pitting at vessel ends. Also for the non-plant oriented readers, it might be useful to make a distinction between the term "membranes" as applied to pit membranes and the plasma membrane of living cells.

Comparison of measured and modeled results appears to distinguish inner and outer domains of the xylem. This is most clear in Fig. 2F, where there are many xylem vessels in outer regions of the xylem that do not appear to be conducting. Readers will wonder why would these presumably newer, wide vessels not conduct? Later in the paper, it is argued that these vessels must not have active hydraulic connections, although this is a speculation.

Setting model pressures independently for each vessel group allowed the model to closely match the measured flows. The paper argues that this is evidence for the existence of pressure gradients across the stems. One could also speculate that this process represents tuning the model in a way that produces the measured results and not necessarily demonstrating that such pressure differences existed in the actual stem. Still, the variation in flow rates around the stem which do not correspond to vessel diameter, suggests that pressures driving flow must vary across the stem.

A general question about MRI flow methods: If we assume the flow profile across an individual vessel (by HP) is a parabolic velocity profile, does the method measure peak flow velocity or average flow velocity?

Fig. 5A shows that there is quite a difference in flows rate among the various xylem elements. Is this because some of them are connected to closer or more actively transpiring leaves? If so, one would certainly expect a non-uniform pressure difference over the xylem.

Although there are confusing aspects to the arguments and presentation, the authors make a compelling case that the lateral connections in a 3D network merit important considerations in the overall picture of xylem hydraulics. Agree that the spatial distribution of pressures and flow need further study.

Specific comments

Flow units: I believe that volume units like m^3/s are more commonly used and easily produced based on density. I assume that mass units more directly arise in the measurements, however.

line 36: Does "semi-permeable membranes" refer to flow through living cells or are "adjacent conduits" directly connected by pits?

line 89: ...and lack of circularity of the vessels...

lines 146, 157: 55 or 135 Pa pressure differences across the stem seem pretty small. It might be important to know how these compared to the axial pressure difference from one end of the stem to the other end. This could be further addressed in the paragraph starting at line 171. The idea of a transverse gradient is within the stem and not exactly across the stem but between individual vessel groups and so perhaps it would be more meaningful expressed as a difference between vessel groups than as a gradient - as Fig 6A shows, there are many differences between vessel groups but not really a gradient across the entire stem.

Fig. 2: Some of the wide vessels in the outer xylem domains do not appear to show flow? This is especially apparent in 2F. Do these vessels correspond to the extra flow in the model image in Fig. 3B. Also in Fig. 2, the usage of terms like dorsal and ventral would seem to be meaningful with animal systems but less so with plants (back, belly??). Perhaps just lower or upper?

Fig. 4: What does it mean to have been "...cropped to inner xylem..."?

Fig 6C: Might be easier to judge in comparison with 6B if the horizontal axis ran from 0 to 10 instead of 0 to 20.

Lines 352-354: This is confusing, one expects that if water flows along a stem through many lateral connections, that the distance traveled is more not less than the simple length of a vessel.

Our responses throughout in blue; line numbers refer to revised manuscript file with revisions marked.

Reviewer #1 (Remarks to the Author):

This paper uses an ingenious approach based on MRI, microCT, and model-fitting to document flow and pressure gradients in a xylem vessel network, resulting in an entirely novel conclusion that narrower vessels contribute more to sap flow than previously predicted. The measurement design was excellent, analyses appropriate, and the conclusions are strongly supported by the data. I expect that some fellow reviewers might complain that the experiment was done with a single grapevine stem, but that single stem yielded a wealth of data and well-supported analyses, so I do not see what replication would have added to this study. Certainly, future research along these lines should follow up by comparing different varieties and species, but it would not be fair to demand such comparisons from this first study.

As this point did indeed come up, we have performed an analysis of additional MRI scans to support the replicability of our main result in *Vitis* (at the end of this document) and added a detailed discussion of the generality of our main result in the manuscript.

I have only one thing to suggest for improving this most important aspect of the manuscript, the redirection of flow to narrower vessels: It might be helpful to readers to add some future research questions in the discussion. For example, what might these findings imply for the effect of embolism in the functioning of the vessel network? What do the findings imply for different degrees of connectivity, different vessel diameter distributions, and different hydraulic integration found in different species? The authors undersell their findings a bit in the discussion by not clearly describing a vision for how their research will affect the field of plant hydraulics. The conclusion that xylem networks are more than the sum of their parts has been reached many times before, so the last sentence of this paper does not do full justice to the counterintuitive and novel findings of this study. The discussion deserves a stronger elevator pitch at the end to make a case for how the findings might affect future thinking in the field.

We have added new text in the Discussion that highlights the implications, and expectations that might be tested in future research, along the lines suggested.

Another important contribution of this study is the conclusive evidence that not all filled vessels are hydraulically active, at least not in this grapevine variety. This is not a new observation, but it contradicts a long-standing assumption in the field of plant hydraulics that non-conductive vessels are mostly embolized and implies that xylem vulnerability curves created by microCT must be interpreted with caution, as not all filled vessels can be assumed to be functional. The authors do not refer to Choat et al. (2010. Measurement of vulnerability to water-stress induced cavitation in grapevine: a comparison of four techniques applied to a long-vesseled species. *Plant, Cell & Environment* 33:1502-1512), which was coauthored by McElrone and Brodersen and in which vessels being filled was equated with vessels being functional for water transport. That paper created strong responses in the plant hydraulics community, and none of these responses are cited either, except for Jacobsen et al. (2018). I do not disagree with the decision to leave out much of that discussion about impacts of long vessels on vulnerability curves, because that is not what this paper is about. However, I think it would be fair and appropriate for the authors to concede that Jacobsen and Pratt (2012. No evidence for an open vessel effect in centrifuge-

based vulnerability curves of a long-vesselled liana (*Vitis vinifera*). *New Phytologist* 194:982-990) were correct when they suggested that many vessels in grapevine are non-functional even when filled and that Choat et al. (2010) failed to take that into account. This should be a clear warning to others who interpret microCT images in terms of xylem vulnerability to embolism. Of course, this has very little bearing on the question whether long vessels create artifacts in hydraulic measurements, and there is no reason to address that question in this paper.

Agreed. We have added this citation (Jacobsen and Pratt 2012) with a remark that they anticipated this result at the appropriate point in the results (line 183).

I also suggest that the recent paper by Pratt and Jacobsen (2018. Identifying which conduits are moving water in woody plants: a new HRCT-based method. *Tree Physiology* 38:1200–1212) should be cited, because it addresses the same general research question as in this study, also using grapevine and microCT. However, it is fair to say that the combination of MRI, microCT, and modeling turned out to be far more powerful than the microCT/staining approach taken by Pratt and Jacobsen (2018).

We have added this citation at the appropriate place in the results (line 205). Indeed, their study helps with the interpretation of our results concerning probable partial xylem immaturity in the outer domain.

Overall, I was very impressed by this study and enjoyed reading the manuscript and thinking about its implications for water transport in plants. The manuscript is in excellent shape, and beyond the couple of suggestions above the only other comment I have is about a typo in line 542, where “fig. 1D” should read “fig. 2D”.

Thank you; fixed.

Reviewer #2 (Remarks to the Author):

In this manuscript the authors present results of MRI (volume) flow rate measurements, X-ray microCT and simulations of xylem anatomy and flow in grapevine. The authors conclude that the MRI results (observed in an intact plant) can only be simulated (on the basis of the anatomy of a 3.3 mm plane) if transverse pressure differences are applied in the simulation. This is due to the vessel network heterogeneity. These transverse pressure gradients redirect flow from wide to narrow vessels, reducing the contribution of the wider vessels to overall sap flow. The relation between xylem anatomy, flow, efficiency and vulnerability of xylem to cavitation is of wide spread interest and is discussed in a large amount of papers. The presented results, if reliable, certainly have important consequences in answering questions about optimal vessel dimensions, xylem conductivity and modelling (dynamics) of flow through plants. So as such the manuscript covers a topic of high interest for a broad audience and is certainly of relevance.

The conclusion is quite contra-intuitive. There is quite some literature available trying to correlate cavitation resistance and hydraulic conductivity and the very complex tissue, vessel network and pit characteristics. The conclusion in this manuscript is based on a combination of three input parameters: (1) volume flow rate (MRI), (2) xylem anatomy of a 3.3 mm slice (microCT) and (3) modelling.

(1) In the manuscript the reliability of the MRI flowmetry cannot really be evaluated. No test results or verification of the MRI results are presented. And only one (?) plant has been measured (true?? I cannot verify from the presented info). MRI on (single) vessel flow has been presented before and has already shown the existence of differences in (local) pressure in individual xylem vessels within the plane imaged (Scheenen et al, Plant Physiol 2010).

Thank you for the reference, we have added a citation to this study (line 184) to support our interpretation of the pressure differences. Indeed, as in that study, a single plant was used in the present manuscript. Nevertheless, as in that study (and in agreement with Reviewer 1), we believe the abundance of information found from this single plant, along with the significant finding therein merits reporting to a broad community. We have performed an analysis of additional MRI scans to support the replicability of our main result in *Vitis* (at the end of this document), noted more explicitly that a single scan is studied here and added a detailed discussion of the generality of our main result in the manuscript.

In that paper also the MRI parameter settings in relation to the dynamic range for the flow measurements (maximal velocity!) have been discussed. Unfortunately, it is impossible on the basis of the presented information in the manuscript to verify if the MRI measurements have been done in the correct way: data on maximal velocities and on small and big delta values of the flow encoding magnetic field gradient are missing. If the dynamic range of the MRI flowmetry was not properly set it can easily explain why larger vessels/high flow rates (which are expected to have the largest velocities) are underestimated (Scheenen et al, 2010).

The parameter settings and maximum possible dynamic range have been added as follows:

“Flow encoding was done stepping the amplitude (G) of the PFGs from $-G_{max}$ to $+G_{max}$, sampling q-space completely and equidistantly in 64 steps, using a flow labeling time (Δ) of 20 ms, a PFG duration (δ) of 3ms and a G_{max} of 0.5 T/m, in this sample allowing the detection of xylem sap flow velocities of up to 5 mm/s.”

The maximum flow velocity that could be detected in the xylem of this sample and with these settings was 5mm/s. In practice, in this sample but also in samples of grapevine of other varieties that were measured before this study, the maximum flow velocities found are about 2.5mm/s. In the current study we chose an unusually high number of flow encoding steps (typical is 32 or 48, here we used 64) to ensure sufficient dynamic range.

A second question related to the MRI experiment is the presence of cavitations. Plant handling to insert the plant in the MRI set up can easily induce cavitations. Xylem in grapevine is quite vulnerable to cavitation, also due to handling/mechanical damage (Jacobsen et al., 2005). Normally it lasts a couple of days to restore flow (if all!) to the maximal values under the given conditions in the MRI system (see e.g. Scheenen et al, 2010, three days for a cucumber plant; Windt et al, 2006). The fact that the MRI water density map (amount of water per volume, not water content!, line 196) shows that the vessels are water-filled does not guaranty that in the xylem system no cavitations are present. Only a slice of 3 mm is imaged by MRI, but the xylem extends over much larger distances. As demonstrated in Scheenen et al, 2010, vessels can be present that are filled but with zero flow, even very close (within the same vascular bundle, so most probably hydraulically connected) to fully functioning vessels. Refilling vessels show a

low velocity (see also Kaufmann et al, . If we consider such situation isolated we indeed have to conclude that local transversal differences in pressure may exist, but the reason is given somewhere else in the xylem system! Large volume vessels are most vulnerable to embolism (see among many others the recent paper of Jacobsen et al, IAWA Journal 40, 2019). Large diameter vessels are also the longest ones. Again, embolism in such vessels can easily explain why flow in the larger vessels (and also why the outer xylem domain) are underestimated, resulting in the presented transverse heterogeneity in pressure differences. Embolism in part of the xylem can even explain strongly reduced flow in xylem regions (Jacobsen et al, 2019; Mrad et al, 2018).

This is a good point, and worth considering, as embolism in the system would block certain pathways and alter flow patterns according to the new network connectivity. The basis of this argument is the possibility that plant handling may have caused air emboli. It is true that plant handling and mechanical damage may affect vessel cavitation resistance. However, this does not necessarily mean that well-watered plants with low xylem tension, upon insertion into the MRI scanner, are likely to incur air emboli due to handling. To substantiate this point: 1) in well-watered tomato plants the xylem was shown to be exceptionally resistant to cavitation by mechanical deformation. Even girdling away the living tissue in the tomato stem or tomato petiole, and severe bending and deformation of the leftover xylem, did not lead to emboli (Van de Wal et al 2017, *New Phytol.* 215). 2) In previous dry down studies on different varieties of grapevine we also did not find any evidence of embolism formation due to handling or insertion in the MRI scanner (Hochberg et al., 2016, *Plant, Cell & Environment* 39; Hochberg et al., 2017, *Plant Phys.* 174). 3) Aside from these published results there are also other, as yet unpublished examples of studies in which long woody vines were inserted into a large MRI scanner (Van As laboratory, Wageningen University, NL). Also in that study, we found that even fairly heavy-handed bending and handling did not cause air emboli in well-watered plants. We therefore find that it is reasonable to assume that the hydraulic conductivity of the plant as it was imaged, was not affected by handling.

Furthermore, our experience with *in vivo* microCT imaging of plants experiencing drought and embolism spread (Brodersen et al. 2012) and recovery from embolism via xylem refilling (Brodersen et al. 2010; 2018; Knipfer et al. 2016; 2017) suggest that the plants used in our experiment should not have had any non-functional vessels due to embolism. The plants were kept in a well-watered state, and embolism in the stem does not typically occur until the plants reach much more negative water potentials. In our experience (Brodersen et al. 2012), the vessels that embolize first are those within close proximity to the pith, and not vessels in the outermost regions as suggested by the reviewer. Embolism clearly begins near the pith and then spreads radially outward toward the bark via intervessel connections. We also find no relationship in our plants between vessel diameter and embolism susceptibility in our previous work (data unpublished).

In our microCT scans of plants in various stages of drought or recovery, vessels are usually in a binary state, they are either filled with water or filled with air. Only under very specific circumstances are they in an intermediate state where the vessels refill. The binary states are easily observable with either microCT or NMR (e.g. Choat et al. 2010; Holbrook et al. 2000) in grapevine. In longitudinal section, the entire vessel is filled with air. Only when the plants are in pots with completely saturated soil and with the lights turned off to prevent transpiration are the plants capable of refilling in our experience. In the current manuscript, our plant was illuminated and most likely transpiring. Measuring transpiration with

a gas exchange system while the plant is in the magnet is not possible, but plants will readily transpire in similar conditions outside the magnet.

In sum, while the possibility that an embolism was present outside of the field of view cannot be excluded, based on our extensive experience, it is not likely to have occurred. It is also very unlikely to have given rise to our observations, with hydraulically inactive vessels around the perimeter rather than the pith and with large vessels in the active xylem domain exhibiting healthy flows—albeit reduced from the HP prediction. As explained below, in response to point (2), external embolism as a cause of the observed transverse pressure gradients is not in the least at odds with our study methodology or interpretation: we expect the forcing to originate to be outside the modelled domain. Indeed, it may contribute somewhat to the heterogeneity of the medium, although the heterogeneity of the un-embolized medium is a sufficient explanation.

On a separate point, with respect to the studies of Jacobsen et al., their “active xylem staining” methods always require the xylem to be opened, introducing potential artifacts into the data. The power of the present study lies precisely in the use of flow profiles from undisturbed xylem. We agree with Reviewer 1 that the combination of methods used here has resulted in a more powerful dataset.

(2) The anatomy was measured in a segment of 3.3 mm by microCT. As has been demonstrated in many papers the hydraulics and flow patterns (and pressure drops) are the result of the total xylem anatomy. Jacobsen et al, 2019, has recently suggested that vessel network traits like vessel diameter and length, number of pits (which tend to be non-evenly distributed over short and long vessels), bordered plates, etc, may be important in developing methods for microCT analysis that are able to accurately predict xylem function. MRI has been measured in a 3 mm slice (same position as the microCT analysed segment) within the intact plant system. Flow (and pressure drops) is the result of the whole network. E.g. embolism and other resistances upstream or downstream the slice will affect the flow observations. I wonder if the MRI measured flow pattern in the slice can be modelled by the anatomy of that slice only! Does it correctly represent the xylem network for modelling the (MRI) observed flow?

The reviewer rightly points out that the pressure drops across the 3mm segment reflect the effects of the xylem network as a whole. That is also the view presented in the manuscript. That is also why the optimization exercise operates on the boundary conditions of our model rather than its parameters or structure: the effect of the xylem network as a whole on the present segment is being discovered. It is necessarily true that *with the right boundary condition* the flow through the 3mm slice can be modelled using only network information from the 3mm slice, as all the external effects simply alter that boundary condition. The present model assumes Hagen-Poiseuille flow in vessel lumina with set resistivity per unit pit membrane area of inter-vessel connections, which is the standard model. Again, the pressure drops inferred this way are surely the result of the network structure outside the scanned segment, but they can be inferred from the flow rates and network xylem observed inside the scanned segment.

In normal practice of hydraulic conductivity and flow measurements, pressure drop and anatomy are related to one and the same segment. In such a setup, flow in a stem segment of grapevine was measured by MRI (Windt, PhD thesis, p 58:). In that situation the largest vessels contributed (by far) the

most, only 31 % of available xylem area was used, and I cannot find any indication of redirection of flow from wide to narrow vessels.

In the cited thesis, total xylem lumen area was compared to the flow conducting area (FCA) and the hypothetical hydraulic conductivity per vessel diameter class was calculated, assuming that all conduits were conductive. In such a study, the redirection of flow from wide to narrow vessels will not show up since no attempt was made to identify which vessels were active, and which ones were not. What was previously missing was the combination of the two imaging methods, which allows for stronger inference than with MRI alone. In the present study, large vessels also provide more flow, consistent with the previous work cited, just not following the fourth-power relation found in the Hagen-Poiseuille equation. The same may be true of the previous work, which lacked the power to make this determination. Interestingly, the wide conduits that were found to be filled but non-conducting in our study, also appear to be present but non-conducting in the grape sample in the PhD thesis (p58, Fig. 7). This may explain the large non-conducting xylem lumen area in this sample.

Some minor questions about the microCT results: 199 solitary vessels, 189 vessels with a single connection, and 103 with more than one (lines 18-139) were reported. What type of connections were included? Pits and/or relays? What resistance values have been used? How were the connections distributed? (see next remark in relation to fig 8).

We have updated the text to include this information, where missing. Please note some of it (e.g. resistance values) are present in the methods section, rather than results, as it reflects outputs of previously published work.

(3) Modelling the dynamics of water flow through plants and trees has attracted continued interest for many years (Tyree, *Tree Physiol* 1988; De Schepper and Steppe, *J Exp Bot* 2010; Zweifel et al, *J Exp Bot* 2007). Such models may be divided into two major classes depending on whether they account for the water storage capacity of stems and leaves, i.e., whether they are dynamic or steady state models, respectively. Water stored in the stems may provide a substantial amount to daily sap flow, but this water is under well-watered conditions replaced by inflows at night (see e.g. review by Landsberg and Waring, *Tree Physiol* 2017). These observations show the importance of storage pools that are connected to the xylem. This has been included in many (hydraulic) models that describe whole plant water (and carbon) flow (see e.g. De Schepper and Steppe, *J Exp Bot* 2010; Zweifel et al, *J Exp Bot* 2007). Xylem vessels cannot be considered as isolated from the storage pools. See also Edwards et al, *Tree Physiol* 1986, who present relation between water potential and relative water content, and relation between relative water conductivity and relative water content.

The idea that stored water contributes significantly to the transpiration stream is the subject of a live controversy in the field of plant hydraulics and is in any case very much a question of appropriate scales. The simple reason is a comparison in the relative sizes in pools and fluxes: if the fluxes greatly exceed potential pools, then the contribution of storage cannot be very significant. Storage likely plays a great role in a giant sequoia, where size and xylem conductivity mean water residence time in xylem may be on the order of days, but hardly so in the case of a 2.5m long grapevine. This is supported by the

literature cited by the reviewer—the review by Landsberg and Waring explicitly talks about capacitance “in large trees” with the prime example of a 54m tall Douglas fir (Čermák et al., 2007). It is true that a model with storage is theoretically strictly better than a model without, *so long as it can be correctly parameterized*. But (a) if the parameters are uncertain, then having zero storage is no better or worse than assuming some non-zero level arbitrarily; (b) if the fluxes greatly exceed potential pools, then the error incurred by assuming steady state is almost certainly negligible. In our case, the total flow through the inner region of the scanned xylem segment is such that its entire volume would be drained in roughly 11 seconds. That’s assuming the entire volume of this domain is just water, available for drainage—clearly an overestimate of the specific potential pools actually present. By contrast, we note empirical studies at the same scale as this one (e.g. Knipfer et al., 2017, *Plant Physiology* 175:1649-1660) that show storage pools theoretically available for water storage in the xylem only rarely get used *in vivo*. We therefore cannot share the view that neglecting storage within the 3mm scanned segment is a problematic aspect of the model.

This view is also supported by considering the differences (scale, tissue) between the present study and the models cited by the reviewer. All three studies cited by the reviewer (Tyree 1988, Zweifel et al. 2007, and De Schepper and Steppe 2010) present models at the whole tree scale, and the latter two over a monthly to yearly time-scale. Zweifel et al. explicitly remark that stem storage is mostly intended for the bark (not sapwood) and De Schepper and Steppe explicitly model a water storage compartment connected only to the *phloem*, not the xylem. Their xylem model is in fact a simplified, bulk version of our own, with connection to phloem. These are reasonable choices at their scales (meters, months), but not ours (microns, seconds). Tyree concludes stem capacitance is an important parameter for soil water depletion but not for the actual water potentials in the plant (p. 211). Moreover, this work is purely theoretical, based on a model sensitivity analysis with non-identifiable and collinear parameters, so it is hard to draw any hard conclusions from it on this particular question. Notably, in their further work, Mel Tyree and his student John Sperry present a number of models at all scales in which xylem capacitance is neglected; much of which work is cited in our manuscript.

As long as transverse pressure differences exist there is a non-equilibrium (pressure, and water potential) situation and radial transport will occur and storage pools cannot be ignored!

It is not clear what is meant by non-equilibrium here. Transverse gradients can arise in steady (equilibrium, in=out) flow. This is illustrated in our fig. 8, but in no way depends on that particular arrangement of resistances. In any porous medium, gradients of pressure will result from the spatial arrangement of conductivities and boundary conditions. If the conductivities are arranged strongly heterogeneously, as in angiosperm xylem, gradients in a direction orthogonal to the main gradient set by boundary conditions will emerge, even in steady state. As the manuscript demonstrates, xylem heterogeneity is itself sufficient for the emergence of these gradients, in steady state and without capacitance.

To a certain amount there will be redistribution within xylem (depending on elasticity), storage pools/cells in different tissues. Such storage pools are not included in the presented model to explain the experimental results, but will certainly affect the outcome and conclusions!

Again, we firmly believe storage in the 3mm segment is negligible and does not affect the validity of our inference of pressure differences across it. Capacitance or storage is also not a good candidate mechanism to explain our main result—flow redirection from wide to narrow vessels. This result has a

much simpler interpretation in the more parsimonious, steady-state model. Its interpretation as being due to storage effects is not very clear: would wider vessels be subject to lower pressure gradients as they are more impacted by storage pools? That seems unlikely, given that their lumina have far greater conductivity, so the conductance/capacitance ratio is higher. In sum, we do not think considering capacitance can improve the manuscript substantially.

In the second scenario to model flows pressure at the top of the sample were set independently in each of the 317 vessel groups (lines 162 – 163). It is not clear to me if at the base the pressure was set equal for all vessel groups. If so, why?? If on the top of the slice (in vivo!) transverse pressure gradients are allowed, why not at the base?? What is essential is the (axial) pressure difference or pressure gradient that drives flow.

We agree that “what is essential is the pressure difference,” not absolute pressure values. Indeed, the absolute values used in the modelling exercise are meaningless, only the differences are constrained by observations. As such, allowing both top and bottom values to change independently would not achieve anything, as only the difference between the two could ever be interpreted. One could either allow the top or the bottom pressure to vary freely and set the other value to achieve the result. Allowing both to vary would result in a non-unique solution set and poor convergence on the pressure difference.

Put another way, our model does not entail a claim the difference arises by the top of the segment having different pressures and not the bottom. It merely entails the claim that the pressure difference across the stem varies transversely, which the reviewer clearly agrees with. The model does not purport to show how the differences arise, but is merely a tool for inference; in this case, for inference of the pressure differences across the segment. When the pressure difference is inferred by standard means here, it shows that wider vessels tend to be subject to lower pressure differences, which is in full accordance with hydrostatic principles, given what we know about the surrounding medium.

If transverse heterogeneous pressure gradients exist over the inspected slice they will exist over the whole length of xylem network and the details of the slice only will not be sufficient to describe flow in the whole system, as argued above. What are the consequences for the conclusions?

Again, we agree that

- (a) the pressure gradients are transversely non-uniform across the entire stem, and
- (b) the pressure differences inferred using the model of the stem segment themselves mostly arise outside the segment.

This in no way means, however, that they cannot be inferred from an accurate xylem network reconstruction and observations of flow on a 3mm slice, nor that “the details of the slice only will not be sufficient to describe flow” in it. We cannot agree with the latter statement, as all flows can always be calculated from medium properties and boundary conditions; no non-local effects are allowable. What our model achieves is to infer the boundary conditions (pressure difference) from the flow observations and medium properties. This can be done regardless of how the boundary conditions arise.

The Ohm’s law analogues presented in fig 8 (and discussed in section Effects of network heterogeneity, lines 246-260 and 288-303) is somehow suggestive. If more (or/and earlier to the left) connections are introduced the outcome will be quite different!

Clearly, fig 8 only shows one, very idealized, configuration of resistances. It is only intended for illustrative purposes, as a demonstration of how network heterogeneity is a sufficient condition for transverse gradients to arise. It is absolutely true that changing the configuration of resistances will result in a different pressure field, with different degrees of transverse non-uniformity, even using the same, uniform boundary conditions. That does not, however, undermine the message of the figure. Rather, it is reinforced: each specific pattern of heterogeneity (connections, resistance values) gives rise to transverse gradients in a given transverse section, despite uniform boundary conditions. So heterogeneity is a sufficient condition for these to arise; the figure cannot address how strong such an effect is in real xylem, but that is the role of the observations and modelling in the study.

It is also notable that what will remain consistent in the figure regardless of arrangement of the resistances is the fact that pressure differences across wider vessel lumina will be smaller. This can even be illustrated by a single lumen in series with a single pit membrane: for equal pit membrane resistance, vessel radius has diminishing returns, which is not an excessively novel theoretical result. What is novel in this study is how that plays out *in vivo*, with an actual significant redistribution of flows away from the wider vessels.

In summary:

- a very interesting manuscript, but essential information is missing (MRI experiment), it is not clear if flow can be modelled on the basis of the anatomy of a 3 mm slice, and it is not clear if the presented model (including scenario 2) is correct under the conditions of transverse pressure gradients.

We have updated the manuscript with the missing information concerning the MRI experiment. We absolutely stand by the validity of the inference of the pressure differences across the 3mm segment from our MRI and microCT data. A steady state model is fully compatible with transverse pressure gradients.

Reviewer #3 (Remarks to the Author):

Review of In-vivo flow redirection from wide to narrow xylem vessels due to pressure gradient heterogeneity

Summary

This work reveals the different in-vivo effects that lead to a deviation of bulk xylem hydraulics from the widely used simplistic addition of Hagen-Poiseuille flows. To do that, this study uses state of the art imaging techniques coupled with a recently developed software to computationally resolve the xylem network. This is then fed to a software that computes water flow through the resolved network. The premise of this paper is interesting and important to the field. The authors make a compelling case of why their study is needed: in the literature, there are no complete theoretical treatments that upscale in-vivo plant vessel, pit membrane, etc. hydraulic functioning to bulk organ water flow. The introduction primes the reader about the goal of the work and sets it up against other work in the field, thus providing a good argument for its originality. Another quality that must be mentioned is the ease of readability of all the figures. The results of this paper, although not surprising, do provide crucial and

previously not available information for scientists to advance the understanding of plant functioning in-vivo. Moreover, the authors did a good job discussing what their analysis of a small segment can and cannot say by enumerating possible external (outside the analyzed domain) factors that might have led to the observed results. These observations will help achieve the much-needed porous medium description of plant xylem in the future.

Major concern

My only major concern is that the observed trends seen in this specific xylem segment might be different or reduced in intensity in different segments and in different angiosperm plant types. The authors do point out that a large portion of the transverse pressure gradients happens because, for example, large vessels are more likely to be connected to a smaller one downstream of the observed segment. If this is true, might we see a less pronounced effect of transverse pressure gradients in, for example, self-supporting ring-porous angiosperms where large vessels in the earlywood are connected to other large vessels? To sum up, 1) how general are the observations seen in this segment? 2) To what extent is this effect particular to vines?

This is a very important point and we have expanded on it in the Discussion (lines 376-400) as well as qualifying certain statements more narrowly (line 370). We had previously avoided discussing it because we think the answer depends on the mechanism behind the emergence of the effect—end wall resistance versus connection of vessels of different diameters in series—which we cannot address with our data. If it is to do with vessels of different diameters, then it is definitely in order to expect flow redirection not to be a very general phenomenon, although it then becomes interesting to note this as a xylem construction constraint that seems to predispose hardwood plants to adopt one of two separate strategies: ring vs diffuse porous wood. If it is to do with wall resistance affecting large vessels more, then the potential generality is greater, as the alternative (increasing pit area with vessel diameter) leads to the safety-efficiency trade-off, regardless of hardwood functional type. What we hope to achieve with this manuscript is to provide evidence that vessel network properties can have a significant influence the way that they perform in planta, and that many of the assumptions that we make need to be reevaluated. The methodology used here can serve as a new framework for understanding exactly the questions that this reviewer raises.

Addressing this point is especially important because neither the title nor the abstract mention that these experiments are done on a vine.

We have rectified this omission in both the title and the abstract.

Line by line comments and minor concerns

1. Line 51: “Volume flow rate” instead of just “flow” so some readers don’t confuse this with velocity. Velocity varies with R to the 2nd power. But the volume flow rate varies with R to the 4th power as stated. Replaced
2. Line 51: “such that a small increase” instead of “and a small increase”. Replaced
3. Line 61: Say “a bulk form of the HP relation...” because even though each vessel might follow the HP

relation if we knew the pressure gradient along each of them, the whole vessel network might not conform to the addition of all HP flows from the individual vessels. Added

4. Line 76: “approximate” and not “determine” Replaced

5. Line 121: “Figure 2B”? Fixed

6. Line 122: “Figure 2A”? Fixed

7. Lines 142-150: One shouldn't really expect that the pressures on top of the segment to be the same along the whole cross-section. This is because the observations are in-vivo so the boundary pressures are set at another level not captured in the study segment (the leaves) as explained by the authors later. The value of that simulation, however, to me, is to show the extent of how external factors might (outside the analyzed domain) contribute to the development of the large transverse pressure gradients. So, it is worth casting the first simulation not as an attempt to model the flow correctly, but as a comparative experiment for the second, more physically accurate, simulation.

Yes, we have rephrased the framing of the baseline scenario to make it clearer that this was our intention.

8. Lines 243-244: In line 138, you say “... 199 are solitary vessels” and this is out of only 491 total vessels in your segment. About 41%. So my gut feeling is that transverse resistances through pit membranes aren't the only story here. It is also about vessel connection patterns upstream and downstream your resolved segment.

We have no doubt that upstream/downstream patterns of connection (as well as boundary conditions on the xylem network as a whole, i.e. leaf and soil water potentials) contribute to the emergence of transverse gradients observed in our sample. Nevertheless, the anisotropy discussed at these lines is a necessary condition for their maintenance (in its absence, the pressures in a transverse plane would quickly equalize in the sample) and would, in a heterogeneous medium with uniform boundary conditions (as in fig. 8), be a sufficient condition for their emergence. Seen from another perspective, the 199 solitary vessels are connected to other vessels in the sample by an infinite ‘transverse resistance’ and this isolation is what allows them to maintain a different pressure level, which arises outside the sample, as pointed out by the reviewer. And these principles remain true outside of the scanned sample, where pit membranes are also mostly oriented transversely, while lumina are mostly oriented axially. This feature of the xylem thus plays a major role in our observations and merits separate attention. We have clarified the text in the sense that we do not mean the transverse connections inside our scanned segment alone give rise to the gradients.

9. Lines 267-268: you repeat the word “gradient” twice (read as the gradient of the gradient), fix this sentence, please. Fixed

10. Line 343-344: I wouldn't say that heterogeneity “rather than” topology affects both local and bulk flows. Vessel geometry and topology throughout the whole plant in fact lead to pressure gradient heterogeneity. These aren't competing effects, but rather one results from the other although not solely.

Both the topology (which resistances are in series, which in parallel) and absolute differences between individual resistance values affect the overall resistance. These are orthogonal effects, as shown in fig. 8, where the latter is changed, without changing the former, and the medium heterogeneity changes as a

result. We have used 'heterogeneity' at this point in the manuscript to mean the latter, but it seems in light of this comment that a better way to put it may be that the overall effect is 'heterogeneity' and it is separately composed of topology and the absolute resistance values, whose effects are not simply additive. We have clarified the text accordingly.

11. Figure 1 caption: Have the vessels been artificially separated during the generation of graphics? How come there are connections where vessels do not touch physically?

In short, yes. This is due to (a) the model formulation, (b) image distortion. What we are showing in 1C is not the segmented images but the idealization. The same 4 vessels are shown in 1B and 1C. Placement of connections is derived from the 3D dataset shows in 1B (where connected vessel lumina come very close to touching at the scan resolution). The idealized conduits in 1C have somewhat reduced radii as compared to the eccentric vessel lumina (geometric mean of major and minor axis), but the main effect stems from an exaggeration in xy plane distances and flattening of z extent for better visualization (a clearer view precisely of those connections).

Reviewer #4 (Remarks to the Author):

General comments

An update to working models might be useful as the authors point out, but it is also worth noting that predictions of flow scaling with the fourth power of conduit radius neglect to consider effects on flow of perforation plates or pitting at vessel ends.

This effect is described at lines 56-59; as noted in the next sentence, it remains an open question whether that effect is distributed uniformly to vessels or not. The present study suggests it may affect wider vessels disproportionately, and this result has been described as counter-intuitive and valuable by three reviewers.

Also for the non-plant oriented readers, it might be useful to make a distinction between the term "membranes" as applied to pit membranes and the plasma membrane of living cells.

We have added 'pit' in front of each use of the word 'membrane'.

Comparison of measured and modeled results appears to distinguish inner and outer domains of the xylem.

The reason for this distinction is described at lines 197-198; with supporting references provided.

The is most clear in Fig. 2F, where there are many xylem vessels in outer regions of the xylem that do not appear to be conducting. Readers will wonder why would these presumably newer, wide vessels not conduct? Later in the paper, it is argued that these vessels must not have active hydraulic connections, although this is a speculation.

We have added references that support the view of immature, water-filled vessels in grapevine. The manuscript attempts to be as agnostic as possible on the cause, though (lines 224-227): immaturity of

the vessels themselves, connection to immature or embolised xylem elements, or absence of transpiring leaves at the terminus of fully formed are all possible reasons for this observation. All of them, however, can be covered by the idea of a lack of active connections to the transpiration stream—if these were present, water would flow. Thus we stand by our original formulation.

Our previously published *in vivo* microCT observations of grapevines in similar conditions (Brodersen et al. 2010; 2012) also show good agreement with the NMR scans here. In general, the outermost vessels can embolize (Brodersen et al. 2013), but often do not. Because of this ambiguity in the microCT scans and the proximity to the cambium, we assume that the outermost vessels are not hydraulically active, either because they are developmentally immature, or because upstream conduits are nonfunctional. Support for this idea is now included in the manuscript in response to Reviewer 1 (see above).

Setting model pressures independently for each vessel group allowed the model to closely match the measured flows. The paper argues that this is evidence for the existence of pressure gradients across the stems. One could also speculate that this process represents tuning the model in a way that produces the measured results and not necessarily demonstrating that such pressure differences existed in the actual stem. Still, the variation in flow rates around the stem which do not correspond to vessel diameter, suggests that pressures driving flow must vary across the stem.

As the reviewer points out, “pressures driving flow must vary across the stem” as simulations with uniform pressures at the top failed to predict flow patterns. The fitting exercise does not demonstrate that real transverse gradients exist in the living stem; the failure of the uniform pressure scenario does, conditional on the model being an accurate representation of flow in vessels. What the fitting exercise does, is to allow us to infer the magnitude of the gradients implied by the data, assuming the validity of the model. Since the model itself is not under dispute (open vessels behave as pipes, pit membranes connect them with non-zero resistance), the object of this comment is unclear. We do not actually tune the model (its parameters are the same in all simulations) but rather ask what boundary conditions allow it to predict the observations. Unless there is a specific question about structural error arising from the model formulation, this exercise is perfectly valid.

A general question about MRI flow methods: If we assume the flow profile across an individual vessel (by HP) is a parabolic velocity profile, does the method measure peak flow velocity or average flow velocity?

The method measures what could be described as the velocity spectrum per pixel (pixel propagator); i.e., it measures how many of the protons within a pixel have traveled how far during a certain flow labeling period. On the basis of the velocity spectrum the following parameters are quantified: flow conducting area (=amount of flowing water), average flow velocity of the flowing water, volume flow (=flow conducting area x average flow velocity), and area (or amount) of stationary water per pixel (if there is any). These three parameters are quantified in a model free fashion for every pixel in a flow map, i.e, no assumptions are made regarding the shape of the velocity profile. For a more complete explanation, please see Scheenen et al., 2000, J. Exp. Bot. 51. In this MS we utilize the total volumetric flow rate in each pixel. These values are then compared to the model’s predictions that correspond to the HP mean velocity multiplied by idealized vessel area.

Fig. 5A shows that there is quite a difference in flows rate among the various xylem elements. Is this

because some of them are connected to closer or more actively transpiring leaves? If so, one would certainly expect a non-uniform pressure difference over the xylem.

This is, of course, exactly right. The boundary conditions on the xylem as a whole are set by the soil and leaves and may be variable on both sides (though in this experiment probably only on the leaf side). As we cannot determine the ultimate source of the transverse non-uniformity on the basis of present data alone, we strive to remain completely agnostic on this issue in the relevant discussion (lines 317-331). At the same time, we demonstrate here how xylem anisotropy and heterogeneity are sufficient to give rise to transverse non-uniformity, regardless of those boundary conditions.

Although there are confusing aspects to the arguments and presentation, the authors make a compelling case that the lateral connections in a 3D network merit important considerations in the overall picture of xylem hydraulics. Agree that the spatial distribution of pressures and flow need further study.

Specific comments

Flow units: I believe that volume units like m^3/s are more commonly used and easily produced based on density. I assume that mass units more directly arise in the measurements, however.

Yes, volume units may be more easily interpretable for most readers. As the reviewer anticipates, we use micrograms of water because they arise in the MRI measurements. At these scales, we prefer not to use an assumed constant density to convert to volume, as this may lead to challenges on a physical basis. A reader that is happy to assume the standard density value may simply read nanoliters for micrograms without any change in the numerical values reported.

line 36: Does "semi-permeable membranes" refer to flow through living cells or are "adjacent conduits" directly connected by pits?

Added "pit"

line 89: ...and lack of circularity of the vessels...

Added "profile", although it has been shown (Lewis, 1992, *AJB* 79(10): 1158-1161) that accounting for vessel shape by deriving separate HP-like equations from the Navier-Stokes relationships does not yield great improvements, that the shape can simply be accounted for by adjusting the 'effective' hydraulic diameter, and that HP provides good predictions for open vessels using geometric mean radius of minor/major axis to account for vessel eccentricity (Zwieniecki et al., 2001).

lines 146, 157: 55 or 135 Pa pressure differences across the stem seem pretty small. It might be important to know how these compared to the axial pressure difference from one end of the stem to the other end. This could be further addressed in the paragraph starting at line 171.

The differences are quite small, but difficult to compare without normalization to length. That is why the paragraph starting at 171 (original manuscript) finds the gradient and discusses it in relation to the overall plant size. This results in a reasonable pressure drop, although we lack data to compare that to. In any case, given that most hydraulic resistance resides outside the xylem, one would expect the actual

axial gradient within the xylem to be less than the overall mean gradient from soil-root to evaporative interfaces, as noted in that paragraph.

The idea of a transverse gradient is within the stem and not exactly across the stem but between individual vessel groups and so perhaps it would be more meaningful expressed as a difference between vessel groups than as a gradient - as Fig 6A shows, there are many differences between vessel groups but not really a gradient across the entire stem.

Indeed, these are local gradients, rather than one gradient across the stem. That is why we talk about "Transverse gradients between adjacent [MRI] pixels", which is also the sharpest meaningful resolution we can achieve, given how the optimization was conducted; sadly we cannot really resolve that further to vessel groups. We use gradients here instead of differences, not because we disagree with the reviewer that differences would be more meaningful in isolation, but because scaling to pixel-pixel distance enables us to do the direct comparison of the local axial and transverse gradients, and highlight that locally, the latter are much stronger. The raw differences are shown in figure 6A and underlying data are available online.

Fig. 2: Some of the wide vessels in the outer xylem domains do not appear to show flow? This is especially apparent in 2F. Do these vessels correspond to the extra flow in the model image in Fig. 3B.

Indeed, the presence of non-functioning vessels is discussed in the corresponding text, with new supporting references on previous observations. Indeed, counting these non-functioning vessels as functioning results in all the spurious flow in the outer domain in fig. 3B, which is why the baseline scenario was repeated for the inner domain only and the remainder of the study focuses on that.

Also in Fig. 2, the usage of terms like dorsal and ventral would seem to be meaningful with animal systems but less so with plants (back, belly??). Perhaps just lower or upper?

For the sake of consistency and clarity, this terminology is in keeping with previous literature (e.g., Brodersen et al, 2013, AJB 100(2):314-321) and commonly used descriptions of xylem specifically in grapevine and sometimes other vines such as tropical lianas. Partly, the issue is, the dorsal zone may end up on the top or bottom.

Fig. 4: What does it mean to have been "...cropped to inner xylem...?"

The full quote is "...to inner xylem domain", which is defined in the caption of fig 2C; the caption has been clarified by reference to the latter. The corresponding simulations only consider the inner domain and so the observations have been cropped accordingly to provide a better visual comparison.

Fig 6C: Might be easier to judge in comparison with 6B if the horizontal axis ran from 0 to 10 instead of 0 to 20.

Thank you for the suggestion. We have adjusted the horizontal axis. For clarity, the means are also indicated on both panels with red lines and their values are reported in the text. We hope this provides additional ease of interpretation to the reader.

Lines 352-354: This is confusing, one expects that if water flows along a stem through many lateral connections, that the distance traveled is more not less than the simple length of a vessel.

It is true that the *total* distance travelled should increase with more crossing via lateral connections, as the total path would be more tortuous. The text, however, is discussing the distance travelled *in each vessel* (line 364-365). The maximum distance that can be travelled in a single vessel is the vessel length. The more frequently a water molecule crosses between vessels, the less distance it will travel *in each one*. This is relevant to applications of Wheeler et al.'s 2005 model if the parameter L' is set to the measured vessel length (as in, e.g. Ooeda et al., 2018, Tree Physiology 38: 223-231). Depending on the questions posed, this may lead to erroneous conclusions if vessel length does not correspond to L' due to the effect described here.

Replicability of the main result

Given the difficulty in accessing and coordinating the necessary instrumentation, we were only able to perform both sets of scans on a single sample at the time of the original experiment and we are unable to re-run the experiment at this time. We are thus constrained to extant MRI scans without paired CT scans, when exploring the potential replicability of the main result: the redirection of flow away from the widest vessels. These scans by themselves have a spatial resolution too low to reliably estimate vessel diameters, which prevents us from pairing flow rates with diameters, to address the question directly.

Instead, we have used each MRI scan to compose a cumulative flow distribution of flow velocities (Fig. R1). This shows what proportion of overall flow occurs at each velocity, cumulated from slowest to fastest. This was done for the scan from the manuscript (black line, Fig. R1 A, B), a more distal internode of the same plant (red line, Fig. R1 A), and a scan of a *V. vinifera* var. Syrah stem of comparable age (red line, Fig. R1 B). We have also used extant microCT scans to generate the same distributions theoretically, following HP plug flow predictions with a uniform pressure difference across the stem, subsequently pixelated into grids at the resolution of MRI scans, for comparison. This was done using vessel diameters from the internode in the manuscript (blue lines, Fig. R1 A) as well as transverse sections through stem internodes of comparable age of three other Syrah plants (cyan, yellow and magenta lines, Fig. R1 B).

Fig. R1: Cumulative flow distributions of velocity, observations and Hagen-Poiseuille predictions for *V. vinifera* var. Cabernet Sauvignon (A) and Syrah (B). Velocities are transformed to effective radius using the HP relation and normalized by the maximum, to eliminate the influence of the absolute magnitude of the assumed pressure difference. The spreads in the predicted distributions reflect uncertainty due to MRI-scale grid placement sampled numerically (n=1000 grids).

Panel A shows that the cumulative flow distribution of the MRI scan from the original manuscript follows an S-shape, while the theoretical prediction using the observed vessel diameter distribution follows a J-shape, consistent with our original findings: more of the observed flow accumulates at lower velocities than predicted. The observation is thus either inconsistent with the vessel diameter profile (which was, however, observed) or with the assumption of uniform pressure difference. This panel also shows that the cumulative flow distribution of velocity of a distal internode of the same plant is consistent with the first internode, rather than the theoretical prediction, assuming it has a similar vessel diameter distribution.

Panel B then shows the variability in the predicted cumulative flow distributions between stems of Syrah of comparable age (due to vessel diameter distributions) and a comparison with the observed distribution of another Syrah stem, which is again more consistent with the observed distribution of the first Cabernet scan than with the theoretical predictions.

These visual inferences are confirmed by pairwise Kolmogorov-Smirnov tests of the cumulative flow distributions, which confidently reject the null hypothesis that the observed distributions and the theoretical distributions are identical, as summarized in Table R1. For this test, the test statistic D is the supremum of the difference between the two distributions, with the sample size in the observation n set to the number of non-zero pixels, the sample size for the prediction m set to the number of vessels in the CT section. The p value found is the one-sided probability that this difference arises between samples of this size from the same distribution.

Table R1: Kolmogorov-Smirnov tests of differences between cumulative velocity profiles

Comparison	Test statistic D	Sample size n	Sample size m	p value
Cab 1 Obs – Cab 1 HP	0.232	553	443	<0.001
Cab 2 Obs – Cab 1 HP	0.214	376	443	<0.001
Syr 4 Obs – Syr 1 HP	0.191	763	450	<0.001
Syr 4 Obs – Syr 2 HP	0.137	763	558	<0.001
Syr 4 Obs – Syr 3 HP	0.237	763	452	<0.001

These results suggest that the two other MRI scans reveal *in vivo* flows consistent with a significant redirection from wider to narrow vessels, as compared to the fourth-power HP prediction. There are, however, two limitations to this analysis. First is the assumption of the same vessel diameter distribution between different transverse sections of the same, or even different plants. This limitation is somewhat mitigated by the comparison with three separate vessel diameter distributions in Syrah and the apparent similarity in maturity and structure of the stem segments involved. The second is that the analysis pushes the limits of the spatial resolution of the MRI observations, as it is impossible to resolve flow rates within individual vessels as a function of radius on this basis alone. This is partly mitigated by

the fact that the Kolmogorov-Smirnov test results remain unchanged if normalized velocity (as opposed to effective radius) distributions are considered.

On balance, this additional analysis thus supports the reproducibility of our main finding in *Vitis vinifera* internodes, even if the evidence is indirect. Regardless, the principles behind the effect are operative across xylem types and contribute to the constraints that result in different construction strategies, as is now discussed in the manuscript. We also believe that while the full analysis was only possible on a section single internode (we have made sure to more explicitly highlight this fact), its results are sufficiently noteworthy to merit reporting.

Reviewers' comments:

Reviewer #2 (Remarks to the Author):

The authors have revised the manuscript according to most of the comments of the reviewers. Doing so, the quality of the ms has been improved. In addition, the authors have presented an extensive response to the comments and questions raised by the reviewers. I agree with most of the response and that here flow in an intact plant is observed in undisturbed xylem.

I still don't agree with two points.

First, in my experience plants have to adapt in the MRI system for a couple of days. In that period it is observed that flow is increasing to a new stable level. It is not right clear what is the reason. In my opinion, emboli cannot be excluded. If emboli are present they will alter the flow patterns. In a dry down study of oak we have observed that the largest diameter vessels embolise first, as demonstrated by combined flow-T2 measurements (see Homan, PhD thesis Wageningen University, 2009).

Second, xylem vessels are clearly not isolated from the surrounding tissues, which can act to a certain amount as storage buffers. This has been observed and reported e.g. in (diurnal) changes in stem diameters in many plants. Even the xylem tissue (surrounding the vessels) can act as storage tissue. Detailed MRI observations can be found in Homan, 2009. Effect of (storage) capacity has also been observed to delay transfer of changes in water content at different positions in the stem of a cucumber plant, and with respect to the change in xylem flow (Van As et al, Biophysical J, 1994). In addition, these water pools can be in contact with air. Water-air interfaces can level off pressure differences. Radial pressure differences will be leveled off, depending on resistances and capacities of the tissues. This all does not mean that stored water contributes significantly to the transpiration stream. I agree that this is hard to model, but anyhow worth to mention in the discussion.

I fully agree that the HP model cannot simply be applied to calculate the contribution of different diameter vessels. In fact, what we need are models based on a comprehensive porous media description of the xylem using Darcy's flow. First attempts have already been published (e.g. Loeffe et al, J. Theor Biol 2007 and others).

Henk Van As

Reviewer #3 (Remarks to the Author):

As mentioned in my previous review, it is my opinion that the insight that this paper puts forward is important to the field. The methodology is novel and the authors document it well. Part of what I hope this paper will bring to the field is to inform better hypotheses on the mechanism of in-vivo water translocation and its vulnerability to environmental change. By understanding better what happens on the smaller scales, the community will better be able to up-scale these processes. My major and minor concerns were answered thoroughly and thoughtfully. The manuscript now better manages reader expectations with a more descriptive title and a more detailed abstract. The added paragraphs in the discussion improve the depth of the main message of the manuscript.

Reviewer #4 (Remarks to the Author):

General comments

The authors have made a good effort to respond to the large number and range of reviewer comments. I leave it of course up to the other reviews to decide if their comments have been adequately addressed. It is always useful to have replication, but I do tend to agree with the authors that a lot of information has been obtained from the single segment utilized in this study.

Regarding the outer ring of vessels that are apparently non-conducting, this is still somewhat of a mystery but the authors speculations as to the cause are not unreasonable.

The authors argue that the lack of ability of the model with the same pressure on all conduits to reflect measured flow rates in vessels indicates that differences exist (transverse pressure gradients) in pressures among vessels at any point along the stem. This is also not an unreasonable conclusion in the sense that there is perhaps no reason to assume that the xylem vessel pressure is the same in all vessels at a particular location along the stem axis. The lack of agreement between model and measurement could indicate as the authors argue that the assumption of no pressure gradient are incorrect. This reviewer's comment about "tuning" the model is not just referring to model parameters, but also the adjustment of boundary conditions. So although I agree that the results from the adjusted model can usefully provide info about the extent of pressure gradients, this is no less an aspect of tuning.

Specific comments:

Line 37: Adding "pit" does resolve this issue about membranes, but I am still left with wondering why "semi-permeable" is still present. In what sense are vessel pit membranes semi-permeable? Passage of air bubbles?

Line 91: The addition of "profile" seems reasonable and adds the aspect of non-circularity. In a general sense, however, I do not agree that only minor differences arise from treating a highly non-circular conduit as a circular conduit of equal area or from approaches using so-called hydraulic diameters. Forms of the HP for non-circular conduits like ellipses or rectangles are easily handled.

I acknowledge the previous use in the literature of terms like dorsal and ventral and so don't find them a major objection. Although, it does get more confusing when stated in the rebuttal that the dorsal zone may be at the top of bottom. So are these anatomically distinct with respect to origin of the tissues (as would be the case with the terms used in animal systems), or is it an effect of top versus bottom? The asymmetry in leaves regarding adaxial and abaxial faces has an anatomical origin. But if a stem has an axial asymmetry, is it up/down (gravity) based? Anyhow, not an issue that needs to be resolved in the present manuscript.

Our replies below in blue, changes to the manuscript are marked.

Reviewer #2 (Remarks to the Author):

The authors have revised the manuscript according to most of the comments of the reviewers. Doing so, the quality of the ms has been improved. In addition, the authors have presented an extensive response to the comments and questions raised by the reviewers. I agree with most of the response and that here flow in an intact plant is observed in undisturbed xylem.

Thank you for your comments, we agree they have significantly improved the manuscript.

I still don't agree with two points.

First, in my experience plants have to adapt in the MRI system for a couple of days. In that period it is observed that flow is increasing to a new stable level. It is not right clear what is the reason. In my opinion, emboli cannot be excluded. If emboli are present they will alter the flow patterns. In a dry down study of oak we have observed that the largest diameter vessels embolise first, as demonstrated by combined flow-T2 measurements (see Homan, PhD thesis Wageningen University, 2009).

We certainly can't disagree that if emboli are present, they will alter the flow patterns. And we also cannot categorically exclude the presence of any emboli on the basis of our measurements alone. We have added a paragraph to recognize and discuss this concern.

We nevertheless do not think emboli are responsible for our observations, for a number of reasons set out in the previous response, which we summarise in the revised manuscript text. To elaborate:

A) Our expectation of no significant embolism due to handling in a well-watered plant is based on a number of studies from our group that showed a lack of embolism while following the same protocols in either microCT (Brodersen et al., *Plant Physiology* 161(4):1820-1829, 2013; Knipfer et al., *PC&E* 38:1503-1513) or MRI (Hochberg et al., *Plant Physiology* 174 (2), 2017; Hochberg et al., *PC&E* 39:1886-1894) instruments. The difference between our expectations and the reviewer's may in part be due to differences among specific NMR instrumentation—some may require more handling (and even pruning) for plant insertion through the narrow bore of the magnet. This was our experience when monitoring grapevine stems for refilling in Choat et al. 2010 with a horizontal bore magnet with an inner diameter of 35mm. In the present study, the NMR device used requires no more handling than would be required for moving the plant to another table in a greenhouse. The magnet is designed specifically for plant use to avoid situations that would damage the stem.

B) Regarding the issue of xylem vulnerability being dependent on vessel diameter, we published data on this topic using microCT in grapevine (Brodersen et al. 2013, Figure S2), which shows no relationship between vessel diameter and susceptibility to embolism. Observations of wide vessels embolising in oak (above) and poplar (Jacobsen et al., *IAWA Journal* 40 (1), 2019: 4–22) provide evidence of a different pattern in other species. However, our experience with grapevine and prior published work from our group shows that in grapevine, embolism starts at the pith and moves radially outward (Brodersen et al., *Plant Physiology* 161: 1820-1829, 2013), and that there is no relation between vessel diameter and likelihood to embolise early, or at the water potentials that the plant in the current study would have

been at during the experiment (Brodersen et al., 2013). Fig. S2 of that article addresses this question specifically (available at <http://www.plantphysiol.org/cgi/content/full/pp.112.212712/DC1>) and shows that large-diameter vessels are not embolized initially, but only progressively, in advanced stages of drying.

A statistical re-analysis of data contained in that figure shows no effect of diameter on embolism status. Using the data for all three plants in that study even in an advanced stage of drying (pressure = -2.3 ± 0.2 MPa, 28% vessels embolised), it is apparent that the largest vessels remain unembolised (Fig. R1). Taking vessel diameter as the predictor variable and embolism status as the response, a generalized linear model with the binomial distribution and a logit linking function shows no effect ($n=2371$ vessels, slope \pm SE = 0.0002 ± 0.0019 , $p=0.934$, $t=0.0827$, $dfe=2369$).

Fig. R1: Diameters and embolism status of ($n=2371$) vessels of three grapevine stems dried to -2.3 ± 0.2 MPa (data from Brodersen et al., 2013). Blue box indicates 25th and 75th percentile, with a red line at the median, black whiskers extent to data within 1.5 times the difference between the 75th and 25th percentile of the box; all points beyond this range are marked individually as outliers.

C) It is difficult to imagine how the usual pattern of embolism in grapevine (innermost-first, not wide-first) would lead to a redirection of flows from wide to narrow vessels, as opposed to radially outward from the pith. If, on the other hand, the widest vessels were to embolise after all, we would expect reductions in their flow far exceeding 25% of their expected flow, as we see here. We would also expect to see some embolized wide vessels clearly in the scans (i.e. large vessels present in CT scans but not water-filled in MRI); this is not the case. Given that vessel lengths are well documented as being quite long in grapevine (sometimes in excess of 1 meter in length), the probability of a vessel ending occurring just outside the field of view that also has an embolized vessel on the other side of the ending, would be quite low. There is also evidence in the literature that large diameter vessels are also the longest vessels,

giving even less support to the hypothesis that embolism outside the field of view would lead to the observed flow patterns. We agree that doesn't exclude the possibility of embolism entirely, but it does make it very unlikely that embolisms in the widest vessels are sufficiently widespread throughout the medium to explain our observations.

D) Finally, hydration status in plant xylem is dynamic in any case. Even if flow in the widest vessels in the scan were slowed down by preferential connection to wider, embolized vessels outside the scan, that would not necessarily make the observation artefactual, but would rather suggest that both xylem heterogeneity itself and its significance to the overall flow pattern increase under water stress.

Second, xylem vessels are clearly not isolated from the surrounding tissues, which can act to a certain amount as storage buffers. This has been observed and reported e.g. in (diurnal) changes in stem diameters in many plants. Even the xylem tissue (surrounding the vessels) can act as storage tissue. Detailed MRI observations can be found in Homan, 2009. Effect of (storage) capacity has also been observed to delay transfer of changes in water content at different positions in the stem of a cucumber plant, and with respect to the change in xylem flow (Van As et al, Biophysical J, 1994). In addition, these water pools can be in contact with air. Water-air interfaces can level off pressure differences. Radial pressure differences will be leveled off, depending on resistances and capacities of the tissues. This all does not mean that stored water contributes significantly to the transpiration stream. I agree that this is hard to model, but anyhow worth to mention in the discussion.

We do not dispute that xylem vessels are hydraulically connected to storage pools, including within the xylem. We have thus added a brief consideration of this mechanism into the manuscript. Even more so than in the case of embolism, above, we strongly believe this does not explain our observations for reasons we have outlined in the previous response.

Diurnal stem shrinking and swelling is commonly thought to be mostly due to hydration status of phloem, not xylem. The two are connected, of course, but at a greater scale (xylem-phloem) than is being investigated here (vessel-vessel differences in water potential), so is unlikely to contribute to the fine-scale gradient heterogeneity we find. Moreover, in grapevine, trunk shrinkage is quite limited and a poor correlate of water status (Intrigliolo and Castel, Irrig. Sci. 26:49-59, 2007).

Our main contention is that the storage component in first-year grapevine xylem is overwhelmingly likely to be negligible as compared to the flow component. Again, a simple comparison of the magnitudes of flow versus volume is the best demonstration of this. Even in the classic Zweifel et al. (Tree Physiology 21:869-877, 2001) study on 6-year old spruce (whose xylem conductance and capacitance points to far likelier use of stored water than grapevine), the stem separately accounts for only 1.1% of daily transpired water, of which the vast majority is envisaged to come from bark, not wood. But it seems to hold also for the cited work on cucumber (Van As et al., JXB, 45(270): 61-67, 1994), where the authors find:

“Sudden changes in light intensity and RH are almost immediately followed by changes in flux and in R_2 . The time dependence of the flux at both positions of the plant stem are found to be in close agreement. The time-dependence of R_2 values, related to the water content of the plant stem tissue [...] is different... The xylem vessels in the stem have a low hydraulic resistance and water capacity, resulting in a nearly simultaneous change of flow rate at any position in the stem. However, R_2 reflects water in the surrounding tissue, with a large contribution of the vacuolar water, which has a high (radial) resistance for transport of water and a large water capacity.

Therefore, the changes in water content as observed by $R2$ are delayed with respect to those of the flow rate...”

Such observations confirm the dominance of the transpiration forcing on water potential (gradients) within vessels: the gradients are set up or relaxed near-instantly following a change in leaf boundary conditions, whereas the storage component equilibrates more slowly, given its higher resistance and capacitance. During transpiration, the main root-leaf gradient is thus the dominant forcing and transverse heterogeneity is set up by the pattern of resistance in the vessel network.

Similarly, in the dissertation by N. Homan, Chapter 4 shows and discusses very interesting radial flow differences and demonstrates that xylem tissue stores water in *Laurus nobilis* L.. The magnitude of this storage, however, is measured in arbitrary units, and only shows a ~3% diurnal fluctuation while flow rate changes by 50-100% between night and day (Fig. 4.6).

More broadly, the effect of any storage pools on water potential (gradients) must be proportional to their effects on flow rates ($Q=K*dP/dx$). So it is not entirely clear what significant effect the reviewer envisages, without a comparable effect on flows. Similarly, the effect of any air-water menisci in xylem storage elements is unclear, as those menisci need not be ‘relaxed’ ($\psi=0$) but may be under tension, as determined by their position in the transpiration stream. Without any guiding observations or evidence, we dare not speculate on such phenomena in the manuscript. We will note here, however, that if water storage pools are in equilibrium with air in vivo, we agree that this should act to dampen transverse (and, indeed, axial) gradients in the xylem (though only in inverse proportion to the resistance separating the storage pool from the adjacent vessel), but then the gross effect of xylem heterogeneity would be even greater than observed here, as we would be seeing the net result of two countervailing effects.

I fully agree that the HP model cannot simply be applied to calculate the contribution of different diameter vessels. In fact, what we need are models based on a comprehensive porous media description of the xylem using Darcy's flow. First attempts have already been published (e.g. Loepfe et al, J. Theor Biol 2007 and others).

We absolutely agree that a porous media perspective on xylem flow is long overdue and we hope our work here contributes to building it up.

Reviewer #3 (Remarks to the Author):

As mentioned in my previous review, it is my opinion that the insight that this paper puts forward is important to the field. The methodology is novel and the authors document it well. Part of what I hope this paper will bring to the field is to inform better hypotheses on the mechanism of in-vivo water translocation and its vulnerability to environmental change. By understanding better what happens on the smaller scales, the community will better be able to up-scale these processes.

My major and minor concerns were answered thoroughly and thoughtfully. The manuscript now better manages reader expectations with a more descriptive title and a more detailed abstract. The added paragraphs in the discussion improve the depth of the main message of the manuscript.

Thank you; we also feel the manuscript has improved thanks to reviewers' comments.

Reviewer #4 (Remarks to the Author):

General comments

The authors have made a good effort to respond to the large number and range of reviewer comments. I leave it of course up to the other reviews to decide if their comments have been adequately addressed. It is always useful to have replication, but I do tend to agree with the authors that a lot of information has been obtained from the single segment utilized in this study.

Regarding the outer ring of vessels that are apparently non-conducting, this is still somewhat of a mystery but the authors' speculations as to the cause are not unreasonable.

Our view of this phenomenon is consistent with the work of Jacobsen et al., 2018, *AJB* 105(2) 142-150, "Functional lifespans of xylem vessels..." and studies reviewed therein. Mature-looking but as yet non-functioning vessels are documented in grapevine and several other species. While there is not full agreement on what they are filled with (gels, or as in this case water) and we can, indeed, only speculate as to what precisely is lacking for them to begin functioning, the idea that this is the last stage of the maturation process is quite well supported.

The authors argue that the lack of ability of the model with the same pressure on all conduits to reflect measured flow rates in vessels indicates that differences exist (transverse pressure gradients) in pressures among vessels at any point along the stem. This is also not an unreasonable conclusion in the sense that there is perhaps no reason to assume that the xylem vessel pressure is the same in all vessels at a particular location along the stem axis. The lack of agreement between model and measurement could indicate as the authors argue that the assumption of no pressure gradient are incorrect. This reviewer's comment about "tuning" the model is not just referring to model parameters, but also the adjustment of boundary conditions. So although I agree that the results from the adjusted model can usefully provide info about the extent of pressure gradients, this is no less an aspect of tuning.

It is certainly true that we used an optimisation approach to arrive at our inference, and in this sense there is an aspect of tuning to our methodology.

Specific comments:

Line 37: Adding "pit" does resolve this issue about membranes, but I am still left with wondering why "semi-permeable" is still present. In what sense are vessel pit membranes semi-permeable? Passage of air bubbles?

Thank you for raising this again. Indeed, we meant that the membrane tries to exclude air while letting water pass. As the wording seems to be more of an obstacle to communication than anything, we've cut this term out to avoid confusing readers.

Line 91: The addition of "profile" seems reasonable and adds the aspect of non-circularity. In a general sense, however, I do not agree that only minor differences arise from treating a highly non-circular conduit as a circular conduit of equal area or from approaches using so-called hydraulic diameters. Forms of the HP for non-circular conduits like ellipses or rectangles are easily handled.

We agree entirely that it would not be too difficult to substitute assumed circular profiles with other simple shapes. It is not clear, though, what would be gained this way, given that vessel shape is not only irregular but also variable: at this scale the problem seems more to do with using a single shape for all vessels, rather than choosing the wrong one. In the ideal case, Navier-Stokes solutions for each particular vessel profile would be used. But such a level of precision is hardly warranted at present, given all the other approximations in current models. Perhaps once the entire 3D network can be modelled with full Navier-Stokes equations, there would be value to getting the profiles right. We are still far from the day when this is observationally and computationally feasible. Meanwhile, it is important to note that it is standard practice to find vessel hydraulic diameter using geometric means of major and minor diameters, resulting in a vessel of slightly reduced area, precisely to account for the irregularity of the real shape.

I acknowledge the previous use in the literature of terms like dorsal and ventral and so don't find them a major objection. Although, it does get more confusing when stated in the rebuttal that the dorsal zone may be at the top of bottom. So are these anatomically distinct with respect to origin of the tissues (as would be the case with the terms used in animal systems), or is it an effect of top versus bottom? The asymmetry in leaves regarding adaxial and abaxial faces has an anatomical origin. But if a stem has an axial asymmetry, is it up/down (gravity) based? Anyhow, not an issue that needs to be resolved in the present manuscript.

You are certainly right that the terms are not in perfect analogy to their use in animal systems. What is being described is that grapevine xylem properties vary periodically in the tangential direction around the stem. Two 'lateral' zones are clearly distinguishable from two other zones, with differences in vessel radius and connectivity. While 'dorsal' and 'ventral' may not be the perfect words to describe the two zones with wider, longer vessels, thought to connect to more distal, distant internodes, we have not found better words as yet.

REVIEWERS' COMMENTS:

Reviewer #2 (Remarks to the Author):

I recognize the authors have adequately handled and responded to the points raised.

It would certainly be interesting to follow the flow and water density behaviour in grapevine during dry down experiments by MRI to follow the embolism formation in vivo. However, that is a new experiment, and not a part of this manuscript.

As demonstrated by Loepfe et al (2007) and others a porous media perspective and the use of Darcy's law to describe xylem flow has already been used. The results presented in this manuscript further stimulate such approach.

Dr. Henk Van As

Our responses below in blue; manuscript changes marked (these were in response to editorial requests only).

REVIEWERS' COMMENTS:

Reviewer #2 (Remarks to the Author):

I recognize the authors have adequately handled and responded to the points raised.

It would certainly be interesting to follow the flow and water density behaviour in grapevine during dry down experiments by MRI to follow the embolism formation in vivo. However, that is a new experiment, and not a part of this manuscript.

As demonstrated by Loepfe et al (2007) and others a porous media perspective and the use of Darcy's law to describe xylem flow has already been used. The results presented in this manuscript further stimulate such approach.

Dr. Henk Van As

Thank you for your comments. We agree that following the hydrodynamics of xylem flow in a drought or even just diurnally would be very valuable. As you note, we were unfortunately only able to observe the hydrostatics of flow here.